

# Water Vapor Transport and its Influence on Water Stable Isotope in Dongting Lake Basin

Xiong Xiao[1], Xinping Zhang[1,2*], Zhuoyong Xiao[1], Zhongli Liu[1], Dizhou Wang[1],

Cicheng Zhang[1], Zhiguo Rao[1], Xinguang He[1,2], Huade Guan[3]

[1] *College of Geographic Science, Hunan Normal University, Changsha 410081, China*

[2] *Key Laboratory of Geospatial Big Data Mining and Applications in Hunan Province,*

*Hunan Normal University, Changsha 410081, China*

[3] *College of Science and Engineering, Flinders University, Adelaide SA 5001, Australia*

**ABSTRACT:** Understanding water vapor sources and transport paths is essential for assessing the water cycle and predicting precipitation accurately. Utilizing water vapor diagnosis and calculations, this study determined the water vapor sources and transport paths leading to precipitation in the Dongting Lake Basin in four seasons (represented by January, April, June, and October). In January, the water vapor generating precipitation originated from the Arabian Peninsula, driven by the southern branch of the westerlies over the southern side of the Tibetan Plateau, along the northern side of the Indian Peninsula through southwest China to reach the Dongting Lake Basin. In April, two transport paths emerged: one aligned closely with the January transport path but the location shifted slightly northward by one degree of latitude, and another was driven by the weak subtropical high over the southwestern Pacific, bringing moist air from the western Pacific via the South China Sea and Indochinese Peninsula. In June, the Dongting precipitation sourced from the northern branch of the South Indian Ocean

---
* Corresponding author. Tel.: +86-13308486020; E-mail address: zxp@hunnu.edu.cn





subtropical high, crossed the equator and transported through various water bodies to
southwestern China, finally reaching the basin. October saw a water vapor transport
path from the western Pacific, crossing the South China Sea, and entering the Dongting
Lake Basin influenced by the East Asian monsoon system. In different seasons, the
variations in water stable isotopes along water vapor transport paths adhered to
Rayleigh fractionation and water balance principles. These findings highlight the
impact of atmospheric circulation on precipitation and isotopes, providing a framework
for understanding water vapor isotope mechanisms and reconstructing past atmospheric
conditions.
**Keywords:** Dongting Lake Basin; Water vapor sources; Transport paths; Precipitation
isotopes; Precipitation amount.
**Significance Statement**
This research explores how water vapor transports and contributes to precipitation in
the Dongting Lake Basin throughout different seasons. Understanding these paths is
crucial because it helps us predict and manage water resources better, which is vital for
agriculture, ecosystems, and communities relying on this water. By identifying the
origins and paths of water vapor, we gain insights into how global climate patterns
influence local weather. This knowledge is not only critical for accurate weather
forecasting but also for preparing for future climate changes. Our findings highlight the
complex interactions between the atmosphere and water cycle, offering a clearer picture
of how seasonal shifts in atmospheric circulation impact regional precipitation patterns.
**1. Introduction**



Diagnosing water vapor sources and analyzing water vapor transport are routine
and foundational tasks, particularly within hydrometeorological services (Gimeno et al.,
2020; Xu et al., 2020). A correct understanding of water vapor sources and transport is
crucial for accurately evaluating the hydrological cycle and effectively predicting
precipitation. For instance, in weather forecasting for the East Asian region, an essential
condition for the occurrence of precipitation is the presence of sufficiently warm and
moist air from low latitudes (Barker, et al., 2015; Tang et al., 2015; Hu et al., 2021).
Moreover, the primary cause of meteorological drought in the East Asian monsoon
region is often attributed to an anomalous decrease in water vapor sourced from the
Bay of Bengal (He et al., 2022; Liu et al., 2023). Additionally, studies have shown that
the abundance of stable isotopes in speleothems is related to monsoon intensity and is
consequently linked to water vapor transport (Rao et al., 2013; 2016; Liang et al., 2020).
Over the past few decades, the employment of diverse mathematical models has
been the crucial approach to track and deduce atmospheric water vapor sources and
transport paths (Gimeno et al., 2020; Xu et al., 2020; Pranindita et al., 2022; Lekshmy
et al., 2022). For instance, Pranindita et al. (2022) employed the water vapor tracking
model WAM-2layers to trace back the water vapor sources during heatwaves in
northern, western, and southern Europe, the reasons for the reduction of the local
precipitation can be attributed to a significant reduction in water vapor supply from the
North Atlantic due to anticyclonic patterns, along with the increased water vapor fluxes
from eastern Eurasia and local regions. Utilizing the Lagrangian model FLEXPART
v9.0, Pérez-Alarcón et al. (2023) identified precipitation water vapor sources associated



with the development of Indian Ocean tropical cyclones. Results showed that the water
vapor sources and transport mechanisms were different during different lifecycle stages
of tropical cyclones. Among numerous methods, the use of HYSPLIT for tracking water
vapor sources is widespread, which employs backward trajectory calculations and
atmospheric wind field information to derive water vapor transport trajectories at given
heights during a precipitation event, making it commonly used for tracing water vapor
during short-duration precipitation events (Draxler and Hess, 1998; Esquivel-
Hernández et al., 2019; Nie and Sun, 2022; Liu et al., 2023). However, due to inherent
model structure and tracking principles, derived water vapor transport paths at different
heights may vary or even be opposite. Moreover, this method cannot ascertain whether
the tracked water vapor indeed causes precipitation, nor can it provide information on
the magnitude of water vapor transport (Wu et al., 2015; Wu et al., 2022; Deng et al.,

2024).

With the continuous improvement of observational techniques and analytical
methods, utilizing reanalysis data to determine the water vapor sources that cause
precipitation has become a common practice (Sun et al., 2011; Hoffmann et al., 2019;
Guo et al., 2019). For instance, Sun et al. (2011) investigated the climatic characteristics
and decadal variations in water vapor transport in Eastern China based on NCEP/NCAR
reanalysis data from 1979 to 2009. The results revealed that the variability in water
vapor transport in the region is attributed to the combined influences of the Indian
summer monsoon and the East Asian summer monsoon. Based on the dataset from
ERA5 and isoGSM2, Xiao et al. (submitted) found a strong positive correlation



between seasonal precipitation and seasonal water vapor budget in the Changsha region.
They noted that southwestward water vapor transport contributes significantly to water
vapor input in all seasons, and only southwestward water vapor flux exhibits a highly
significant positive correlation with regional precipitation amount. Although water
vapor input from the northwest direction exists, there is no correlation between water
vapor transport in that direction and water vapor budget or precipitation amount, while
it even shows a negative correlation in some cases. Since the direction of water vapor
transport has an important influence on regional precipitation, it is necessary to reveal
the influences of atmospheric circulation such as the water vapor source regions and
water vapor transport paths, which determine the water vapor transport direction.
Water vapor transport controlled by atmospheric circulation not only determines
precipitation events but also directly influences the precipitation isotopes, thus
analyzing the water vapor sources and water vapor transport paths, as well as their
influences on stable isotopes under different seasons, can elucidate the mechanisms
influencing the atmospheric stable isotopes (Zhou et al., 2019; Dahinden et al., 2021;
Zhan et al., 2023). For instance, Risi et al. (2010) conducted an analysis of water vapor
and precipitation isotopes in the Sahelian region by combining water vapor budget and
water vapor transport calculations, revealing that the isotopic composition of
precipitation and atmospheric water vapor in the region is controlled by the intensity of
air dehydration and changes in convection. Similarly, Sengupta et al. (2006) quantified
the influences of different water vapor source regions on precipitation in the northern
Indian monsoon region, finding that the isotopic composition of precipitation in the





region is influenced by changes in water vapor source and atmospheric circulations over
India. Moreover, Zhou et al. (2019) separately computed the correlations between $\delta^{18}O$
values of precipitation at different sites and found that during the prevalence of the
summer monsoon (April to September) and winter monsoon (October to March), the
key upstream regions influencing the precipitation isotopes in the Dongting Lake Basin
located in south-central China are the Bay of Bengal and southwestern China,
respectively. However, as the critical regions influencing regional precipitation isotopes
may not necessarily be the water vapor source regions, these studies are yet to
definitively determine the water vapor source regions and water vapor transport paths.

Existing studies indicate that in the East Asian monsoon region, including the

Dongting Lake Basin, differences in the water vapor sources and transport direction
during different seasons are the primary drivers of seasonal variations in precipitation
isotopes (Araguás-Araguás et al., 1998; Zhang et al., 2016; Wei et al., 2018; Chiang et
al., 2020). Typically, during the summer monsoon, prevailing southeast or southwest
winds dominate the East Asian monsoon region, with water vapor for precipitation
originating from low-latitude oceans (Barker, et al., 2015; Wu et al., 2015; Tang et al.,
2015), while precipitation isotopes are significantly depleted influenced by intense
rainout effects along the water vapor transport paths during this period (Zhou et al.,
2019; Wu et al., 2022). Conversely, during the winter monsoon, northwest or northeast
winds prevail in the East Asian monsoon region, by simple deduction, the precipitation
isotopes should be more enriched if water vapor for precipitation is carried by westerlies
or originates from the evaporation of inland regions (e.g., Liu et al., 2011; Wu et al.,



2015; Shi et al., 2021). However, both actual observations from the Global Network of
Isotopes in Precipitation (GNIP) and simulations from isotope-enabled General
Circulation Models (isoGCMs) consistently demonstrate that, whether during the
summer or winter monsoon, the spatial distribution of precipitation isotopes in the East
Asian monsoon region exhibits significant latitudinal and continental effects—that is,
the precipitation isotopes become more depleted with the increases of latitude or
distance from the ocean (Feng et al., 2009; Zhang et al., 2012; Zhang et al., 2016).
Consequently, the observed water vapor transport during the summer monsoon aligns
with the spatial distribution of precipitation isotopes under the influence of latitudinal
and continental effects and is consistent with the Rayleigh distillation principle for
water stable isotopes, however, water vapor transport during the winter monsoon does
not follow the above spatial distribution and Rayleigh distillation principle (Tang et al.,
2015; Zhou et al., 2019; Wu et al., 2022).

Based on the understanding outlined above, a thorough investigation into the

seasonal variations in water vapor sources and transport paths for precipitation amount
and isotopes in the East Asian monsoon region is necessary, which may provide
significant benefit for accurately understanding regional hydrological mechanisms and
elucidating regional climate characteristics. Focusing on the Dongting Lake Basin
within the East Asian monsoon area, and drawing upon fundamental theories of
meteorology, water vapor diagnostics, and water vapor calculations, this study aims to
(1) identify the water vapor sources and transport paths contributing to the Dongting
Lake Basin; (2) analyze the variations in meteorological factors and water stable



isotopes along the water vapor transport paths; and (3) reveal the mechanisms by which
water vapor sources and transport paths in the monsoon region influencing precipitation
amounts and isotopes.
**2. Methods and materials**
**2.1 Study site**
Dongting Lake Basin, situated in the south-central region of China (Fig. 1), is a
basin characterized by a subtropical monsoon climate, marked by distinct four seasons
and moderate humidity. Winters are moist and cold, while summers are warm and moist.
Based on historical meteorological data from 1960 to 2017, the Dongting Lake Basin
experiences an average annual precipitation of 1375.6.0 mm. During the colder months
(October to March of the following year), precipitation is relatively low due to the
influence of continental air masses. However, from late April onward, influenced by
maritime monsoons, precipitation increases significantly, accompanied by a notable
rise in temperature, with precipitation predominantly occurring from April to June (Liu
et al., 2023; Xiao et al., 2024).

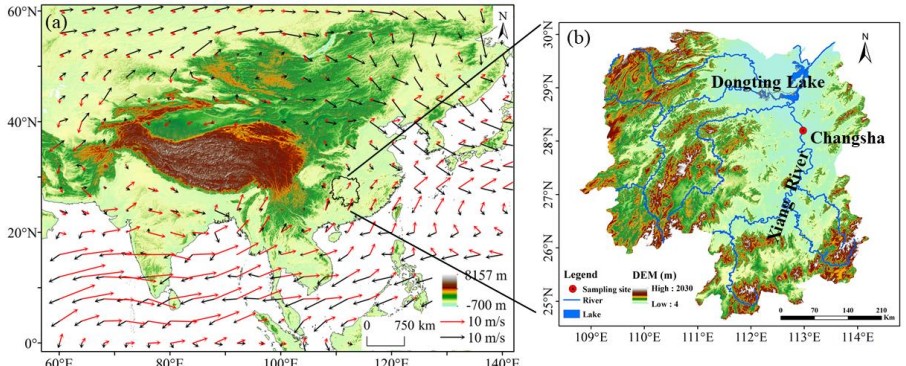


Fig. 1 Map showing the location of the Dongting Lake Basin, and the Changsha





sampling site in the East Asian Monsoon Region. Note that the black arrow and red
arrow in subplot (a) represent the average wind field at 850 hPa in January and June,

respectively.

The prevailing wind refers to the wind or wind direction that appears the most
frequently in a region during a specific period. Its occurrence is closely related to the
atmospheric circulation of the region. In the East Asian monsoon region, which includes
the Dongting Lake Basin, the strong cold high-pressure system influences the winter
season, resulting in prevailing northerly winds near the surface, with northwesterly
winds prevailing in the basin as shown by the average wind field at the 850 hPa in
January, i.e. the black arrow in Fig. 1a. In the summer, influenced by the Western Pacific
Subtropical High and the Indian Low, the near-surface winds are predominantly
southerly in the East Asian monsoon region, with southwesterly winds prevailing in the
Dongting Lake Basin as shown by the average wind field at the 850 hPa in June, i.e.
the red arrow in Fig. 1a. Positioned at the convergence of the prevailing northerly winds,
prevailing southerly winds, and westerly winds, the Dongting Lake Basin experiences
complex precipitation processes and different precipitation amounts in different seasons
and water vapor transport directions. This complexity results in high variability in the
precipitation isotope dynamics (Zhou et al., 2019; Xiao et al., 2024).
**2.2 Water samples collection and analysis**
From January 1, 2010 to December 31, 2022, precipitation sample sampling has
been conducted at the Meteorological Garden of Hunan Normal University in Changsha
(28°11'N, 112°56'E). The sampling protocol followed the meteorological observation





standards of China's meteorological departments, with samples collected at 08:00 and
20:00 Beijing time on precipitation days. Liquid precipitation was directly collected in
sealed 30 ml polyethylene bottles after measuring the precipitation amount, while solid
precipitation was first collected in air-tight plastic bags, then measured for meltwater
volume after natural melting, and transferred to the same size polyethylene bottles. All
the collected water samples were stored in a refrigerator at 0°C before testing.

Precipitation sample analysis from 2010 to 2013 was conducted using a Liquid

Water Isotope Analyzer (DLT-100, Model: 908-0008) from Los Gatos Research, USA;
subsequently, a new generation Liquid and Gas Dual-Mode Stable Isotope Analyzer
(IWA-35EP, Model: 912-0026-1000) from the same company was used from 2014 to
2022. The oxygen and hydrogen stable isotope ratio in the water samples were
expressed in per mil (‰) deviations relative to the Vienna Standard Mean Ocean Water
(V-SMOW), calculated using the equation:
$$\delta^2H \text{ or } \delta^{18}O = \left[ {R_s}/{R_{\text{V-SMOW}}} - 1 \right] \times 1000 \tag{1}$$

In the equation, $R_s$ and $R_{\text{V-SMOW}}$ represent the oxygen (or hydrogen) stable isotope ratios
$^{18}O/^{16}O$ (or $^2H/^1H$) in the water sample and in Vienna Standard Mean Ocean Water (V-
SMOW), respectively. The testing precision averaged $\delta^{18}O \leq 0.3‰$ and $\delta^2H \leq 2‰$
during 2010-2013, and $\delta^{18}O \leq 0.2‰$ and $\delta^2H \leq 0.6‰$ during 2014-2022. If there were
two precipitation samples in one day, the precipitation stable isotope values for that day
were represented by the volume-weighted average. In total, 1668 precipitation days'
$\delta^{18}O$ ($\delta^2H$) data were obtained over the past 13 years.
**2.3 Ancillary data**



ERA5, produced and released by the European Centre for Medium-Range Weather
Forecasts (ECMWF), is the fifth-generation global atmospheric reanalysis data product
from the center. Compared to its predecessor ERA-Interim, ERA5 incorporates a state-
of-the-art integrated forecasting system, integrates more historical observational data,
and reprocesses a large amount of assimilation data, resulting in significantly improved
accuracy (Albergel et al., 2018; Hoffmann et al., 2019). Additionally, ERA5 features
substantial improvements in temporal and spatial resolution. The temporal resolution
has increased from 6 hours in ERA-Interim to 1 hour, while the horizontal resolution
has improved from 79 km to 31 km, and the highest vertical extension reaches 0.01 hPa
altitude. These enhancements enable ERA5 to capture finer atmospheric details.
Moreover, the number of variables provided by ERA5 has increased from over 100 in
ERA-Interim to the current 240, and the data release delay has been reduced from 2-3
months in ERA-Interim to 5 days (Albergel et al., 2018; Hoffmann et al., 2019). The
reanalysis data used in this study include surface pressure, specific humidity at
1000/850/700/600/500/400/300 hPa, altitudinal wind, and meridional wind. The
horizontal resolution is 1°×1°, with a temporal step of 1 hour. This dataset was used to
calculate the vertical integral of water vapor fluxes into a specified region, introduced
in section 2.4.
Since the fractionation process of water stable isotopes in the atmosphere cannot
be directly observed, analyzing the variations of atmospheric stable isotopes requires
the application of stable isotopes fractionation theory along with the fundamental
principles and methods of meteorology. In terms of research methods, the introduction



of atmospheric circulation models for water stable isotope cycling, such as isoGCMs,
provides a unique and effective tool. Among numerous isoGCMs, isoGSM (Isotope-
incorporated Global Spectral Model) exhibits relatively good simulation performance
in the East Asian region (Zhang et al., 2020; Kathayat et al., 2021). isoGSM is a stable
isotope GCM developed by Yoshimura et al. (2008), which integrated water isotope
cycling and fractionation processes into the Global Spectral Model at the Scripps
Experimental Climate Prediction Center. The driving factors include sea surface
temperature, sea ice, and temperature and horizontal wind fields in 28 vertical layers.
This model addresses the Gibbs phenomenon in atmospheric circulation models and
performs better in simulating water vapor transport processes in arid and high-altitude
regions (Yoshimura et al. 2008; Bong et al., 2024). The second-generation isoGSM2
has a higher temporal and spatial resolution in simulating water vapor and precipitation
isotopes compared to the first-generation (Chiang et al., 2020). It utilizes the NECP-R2
(National Centers for Environmental Prediction Reanalysis 2) reanalysis dataset and
abandons the NDSL (Non-iteration Dimensional-split Semi-Lagrangian) advection
scheme used in the previous generation. By dynamically correcting the model output
using reanalysis data, isoGSM2's simulation results are closer to actual atmospheric
conditions, thereby improving the accuracy of water vapor and precipitation isotope
simulations (Bong et al., 2024).

The water stable isotope simulation data used in this study are from isoGSM2

(January 1979 to December 2017, totaling 468 months), including monthly
precipitation amount ($P$), stable isotopes ($\delta^2$H and $\delta^{18}$O) in the precipitation and vertical





integral of water vapor ($\delta^2H_v$, $\delta^{18}O_v$, $\delta^2H_p$, and $\delta^{18}O_p$), and the calculated deuterium
excess in water vapor and in precipitation (Ex_$d_v$ and Ex_$d_p$). The spatial scale ranges
from 30°S to 70°N and 0° to 280°E, with a horizontal resolution of 1°×1° (Chiang et
al., 2020; Liu et al., 2023).
**2.4 Model Analysis**

The water vapor transport flux serves as a metric for both the magnitude and

direction of water vapor transport, representing the mass of water vapor passing through
a unit cross-section per unit of time (Sun et al., 2011). The specific calculation equation
is as follows:
$$Q = \frac{1}{g} \int_{P_t}^{P_s} Vq\mathrm{d}p \qquad (2)$$
Where the meridional component $Q_\lambda$ and the latitudinal component $Q_\varphi$ of the water
vapor transport flux are given by:
$$Q_\lambda = \frac{1}{g} \int_{P_t}^{P_s} uq\mathrm{d}p \qquad (3)$$
$$Q_\varphi = \frac{1}{g} \int_{P_t}^{P_s} vq\mathrm{d}p \qquad (4)$$
Here, $Q$ represents the vertical integral of water vapor flux (kg·m⁻¹·s⁻¹), including the
meridional component $Q_\lambda$ and the latitudinal component $Q_\varphi$. $V$ denotes the vector wind
speed (m·s⁻¹), including the latitudinal wind speed ($v$) and meridional wind ($u$), $q$
represents specific humidity (kg·kg⁻¹), g is the acceleration due to gravity (m·s⁻²), $p_s$ is
the lower boundary pressure (hPa), and $p_t$ is the upper boundary pressure (hPa). In the
actual atmosphere, water vapor content above 300 hPa is minimal, thus $p_t$ is set to 300
hPa when calculating the vertical integral of water vapor flux through the entire



atmospheric column.
**3. Results**
**3.1 Seasonal Variation Characteristics of Precipitation Isotopes in the Changsha**
**Region**

The monthly weighted average and total monthly calculations were performed on

the daily $\delta^{18}O_p$, daily Ex_$d_p$, and daily $P$ collected from the Hunan Normal University
and the Changsha National Meteorological Reference Station, yielding the seasonal
variations of multi-year monthly weighted average $\delta^{18}O_p$, monthly weighted average
Ex_$d_p$, and monthly average $P$ in the Changsha region (Fig. 2a). The $\delta^{18}O_p$, Ex_$d_p$, and
$P$ in Changsha exhibited significant seasonal variations—that is, the maximum value
of $\delta^{18}O_p$ appeared in March and April, both at −3.57‰, but did not correspond to the
months with the lowest precipitation amounts. The three lowest values of $\delta^{18}O_p$
occurred in July, August, and September, respectively at −9.45‰, −8.93‰, and
−9.42‰, with a simple arithmetic average of −9.27‰, which also did not correspond
to the months with the highest precipitation amounts. The maximum value of Ex_$d_p$
(20.05‰) appeared in January, and the minimum value (9.29‰) appeared in August,
both of which were months with relatively low precipitation. Due to these significant
differences in the phases of precipitation isotopes and amounts, it is apparent that
explaining the variations in local precipitation stable isotopes solely based on the
seasonal variations in local precipitation amounts is insufficient.



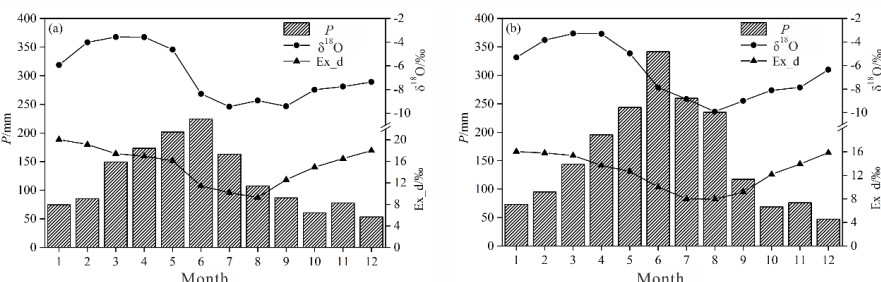


Fig. 2 Comparisons between seasonal variations of precipitation $\delta^{18}O$ ($\delta^{18}O_p$),
precipitation excess deuterium (Ex_$d_p$), and precipitation amount ($P$) measured at the
Changsha station (a) and simulated by isoGSM2 or driven from the RA5 reanalysis

dataset at the corresponding grid (b).

Monthly weighted average calculation was performed on the monthly $\delta^{18}O_p$ and

Ex_$d_p$ simulated by isoGSM2 at the Changsha grid, and the monthly average
calculation was performed on the $P$ from ERA5, yielding the seasonal variations of
simulated monthly weighted average $\delta^{18}O_p$ and Ex_$d_p$ and ERA5 monthly average $P$ at
the Changsha grid (Fig. 2b). The simulated and calculated $\delta^{18}O_p$, Ex_$d_p$, and $P$ in
Changsha all effectively reproduced the seasonal variations of the corresponding
observations. The root mean square errors (RMSE) between simulated and observed
values were 0.54‰, 2.78‰, and 59.7 mm, respectively. Corresponding to the observed
seasonal variations, the two maximum values of the simulated $\delta^{18}O_p$ occurred in March
and April, at −3.29‰ and −3.31‰, respectively, with very small differences from the
observed values. The three lowest values of simulated $\delta^{18}O_p$ also occurred in July,
August, and September, at −8.84‰, −9.92‰, and −9.00‰, respectively, with a simple
arithmetic average of −9.25‰, which was consistent with the observed values. The
maximum value of simulated Ex_$d_p$ (16.05‰) appeared in January, and the minimum





value (7.97‰) appeared in August (Fig. 2b), both consistent with the observed values
(Fig. 2a). These comparisons indicated that isoGSM2 exhibited strong capabilities in
simulating the spatial distribution and temporal variations of atmospheric water stable
isotopes.
To analyze the seasonal variation in the atmospheric water vapor transport and its
influences on the regional precipitation isotopes, and taking into account the hydro-
climatic characteristics of the study region (Fig. 2), four representative months
including January, April, June, and October were selected as the study seasons. Among
these representative months, January in the Changsha region represents winter,
characterized by the lowest temperatures and relatively low precipitation throughout
the year. April signifies spring, with rapidly increasing precipitation amounts and
frequent fluctuations between warm and cold air masses. June represents the peak of
the summer monsoon season, with the highest monthly precipitation amount of the year.
October represents autumn, characterized by clear and cool weather and the second-
lowest precipitation throughout the year under the influence of the West Pacific
Subtropical High.
**3.2 Water Vapor Transport in the Dongting Lake Basin in Different Seasons**
**3.2.1 Average Water Vapor Transport Path in the Dongting Lake Basin in January**
Based on the ERA5 reanalysis data, we calculated and plotted the spatial
distribution of the 500 hPa average geopotential height ($H_{500}$) and average $Q$ (Fig. 3a),
multi-year average $P$ (Fig. 3b) in January. Moreover, based on the isoGSM2 simulation
data, we plotted the spatial distributions of the average $\delta^{18}O_v$ (Fig. 3c), $\delta^{18}O_p$ (Fig. 3d),



Ex_d$_v$ (Fig. 3e), and Ex_d$_p$ (Fig. 3f) in January.

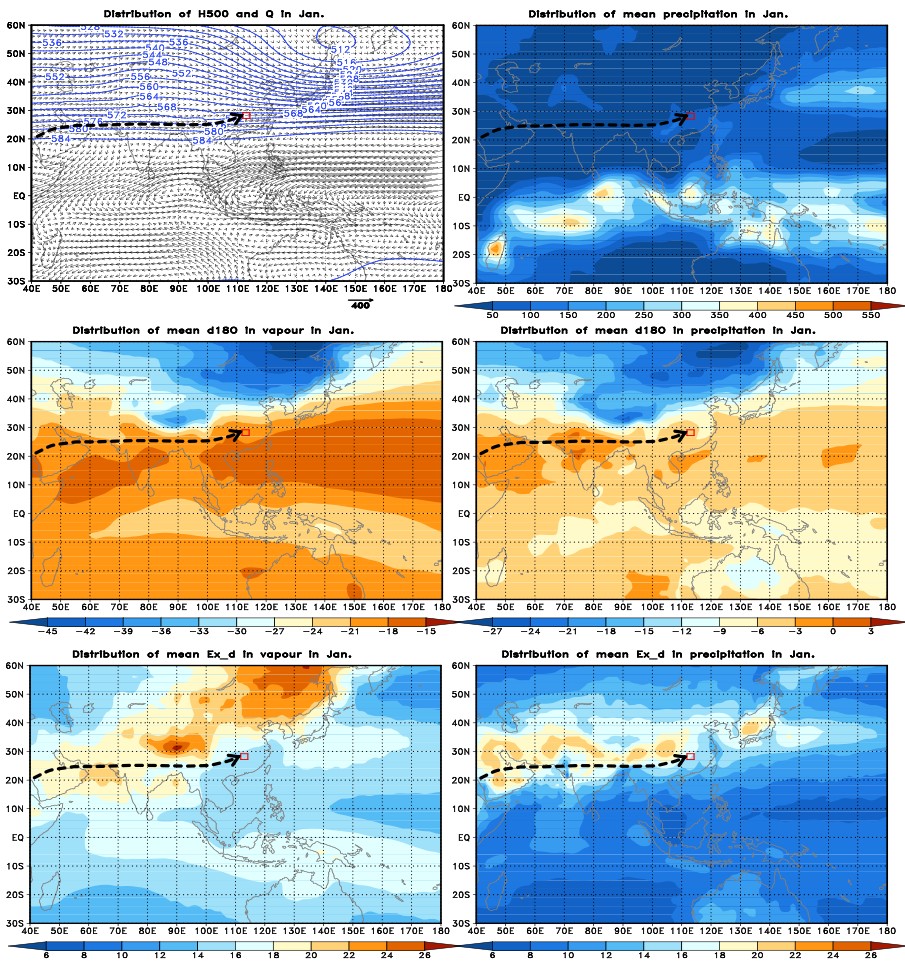


Fig. 3 Mean vapor transport path to the Dongting Lake Basin and the spatial
distributions of $Q$ with $H_{500}$ (a), $P$ (b), $\delta^{18}O_v$ (c), $\delta^{18}O_p$ (d), Ex_d$_v$ (e), and Ex_d$_p$ (f) in
January. $Q$, $H_{500}$, $P$, $\delta^{18}O_v$, $\delta^{18}O_p$, Ex_d$_v$, and Ex_d$_p$ represent the vertical integral of
water vapor flux, 500 hPa average geopotential height, precipitation amount, $\delta^{18}O$ in
atmospheric water vapor and precipitation, and deuterium excess in atmospheric

water vapor and precipitation, respectively, hereinafter the same.

At the $H_{500}$ field of East Asia, the deep East Asian Trough was stably located along



the East coast, and the strong Ural Ridge in the mid-high latitudes of 70°E to 90°E (Fig.
3a). Influenced by the northwest airflow behind the trough and ahead of the ridge, the
northwest winds prevailed in most parts of East Asia. In the Dongting Lake Basin
(highlighted in the red box in Fig. 3), influenced by the middle-latitude westerly belt,
water vapor transport mainly was from west to east. Under the control of the cold
continental high, precipitation was relatively low over the entire East Asia and South
Asia, while the regions with high $P$ values were mainly distributed in the equatorial
convergence zone and the North Pacific located ahead of the East Asian Trough (Fig.
3b). Unlike most regions of the East Asian continent, the Dongting Lake Basin lied on
a wet tongue, benefiting from the Southwest Vortex in the eastern Tibetan Plateau (Lai
et al., 2023; Huang and Li, 2023).

The $\delta^{18}O_v$ and $\delta^{18}O_p$ in January exhibited significant continental effects (Figs. 3c

and 3d). Under the control of continental cold air masses, the centers of minimum $\delta^{18}O_v$
and $\delta^{18}O_p$ values were located in the mid-high latitudes of Eastern Siberia. Due to the
influence of topography, the $\delta^{18}O_v$ tended to be negative over the Tibetan Plateau. These
two low-value regions correspond to the cold pole of Eurasia and the Earth's third pole,
respectively. Regions enriched in atmospheric water isotopes were mainly distributed
over vast oceans and Western Asia. In the equatorial convergence zone, due to the
rainout effects, both water vapor isotopes and precipitation isotopes were depleted to
some extent. Along with the surrounding the Dongting Lake Basin, the abundance of
water vapor isotopes and precipitation isotopes in the Dongting Lake Basin were
comparable to those of the middle-low latitude oceans in January.





The spatial distributions of the $Ex\_d_v$ and $Ex\_d_p$ exhibited the characteristics of
low in the ocean and high in the land, but the regions where their maximum values
occurred did not completely correspond (Figs. 3e and 3f). The maximum value of the
$Ex\_d_v$ mainly appeared in Eastern Siberia and the Tibetan Plateau, showing a
meridional distribution from northeast to southwest, while the high values of the $Ex\_d_p$
mainly occurred in mid-latitude inland regions, showing a latitudinal distribution from
west to east. The $Ex\_d_v$ and $Ex\_d_p$ values in the Dongting Lake Basin lay exactly in the
transition region from low to high values. Typically, the $Ex\_d_p$ largely depended on the
$Ex\_d_v$, but processes such as condensation in clouds, secondary evaporation below
clouds, evaporation from underlying surfaces, and the exchange and diffusion of water
vapor isotopes could cause precipitation isotopes to deviate to varying degrees from
atmospheric water vapor isotopes (Zhang et al., 2016).
In the $Q$ field (Fig. 3a), a vector interpolation method was applied regarding the
Changsha site as the endpoint, as well as based on the drawing of streamlines, the
average water vapor transport path in January was obtained (black arrow lines in Fig.
3). This transport path originated near the Arabian Peninsula. Driven by the southern
branch of the westerly stream jet on the southern side of the Tibetan Plateau, water
vapor transported along the southern side of the Tibetan Plateau, passed through
southwestern China via the northern part of the Indian Peninsula, and reached the
Dongting Lake Basin. It can be seen that this water vapor transport path was not
consistent with the prevailing wind direction in January as shown in Fig. 1a. Six series
of factors at the grid points along the water vapor transport path were derived from each



factor field in January, including the variations of $Q$, $P$, $\delta^{18}O_v$, $\delta^{18}O_p$, Ex_d$_v$, and Ex_d$_p$.
As shown in Fig. 4, both the $Q$ and $P$ were relatively low, while increased to some
extent due to the converging effect of Southwest Vortex after entering the Dongting
Lake Basin (Figs. 4a and 4b). Under the weak atmospheric meridional disturbances in
January, the changes in $\delta^{18}O_v$ and Ex_d$_v$ were minor, fluctuating slightly around $-19.06$‰
and 18.16‰, respectively (Figs. 4c and 4e). Due to the low precipitation amount, the
$\delta^{18}O_p$ values were relatively positive in the first half of the water vapor transport path,
and then became more negative in the latter half with the enhanced rainout effect, while
the Ex_d$_p$ became more positive (Figs. 4d and 4f).

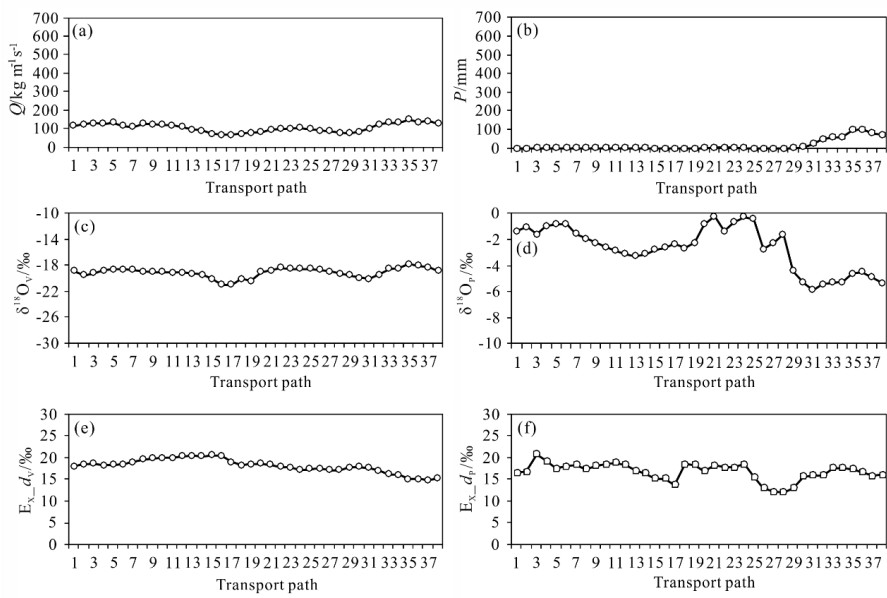


Fig. 4 Mean variations of $Q$ (a), $P$ (b), $\delta^{18}O_v$ (c), $\delta^{18}O_p$ (d), Ex_d$_v$ (e), and Ex_d$_p$ (f)
along the vapor transport path in January
**3.2.2 Average Water Vapor Transport Path in the Dongting Lake Basin in April**
Based on the ERA5 reanalysis data and the isoGSM2 simulation data, the spatial



distributions of $H_{500}$, $Q$, $P$, $\delta^{18}O_v$, $\delta^{18}O_p$, Ex_$d_v$, and Ex_$d_p$ were calculated and plotted
in April (Fig. 5). At the $H_{500}$ field (Fig. 5a), the East Asian Trough and Ural Ridge were
still present in the mid-to-high latitudes. The East Asian mid-to-high latitude regions
were still influenced by the winter monsoon, while its intensity was significantly
weakened. In the mid-to-low latitudes, the Western Pacific subtropical high has
strengthened and expanded northward, with the ridge line located approximately near
15°N. A shallow trough appeared in the northern part of the Bay of Bengal, indicating
the beginning of tropical systems influencing the mid-to-low latitude regions of East
Asia. In the Dongting Lake Basin, influenced by the westerlies and low-latitude
atmospheric systems, the water vapor transport shifted to the domination by the
southwestward direction. Most continental regions of East Asia and South Asia
experienced a certain degree of precipitation increase, with the rainy band caused by
the intertropical convergence zone, previously located in the Southern Hemisphere,
moving to the Northern Hemisphere (Fig. 5b). The Dongting Lake Basin also entered
the spring flood season in April, while the Changsha region was situated in a center
with an above-average spring precipitation amount compared to the surrounding
regions in this period (Fig. 5b).



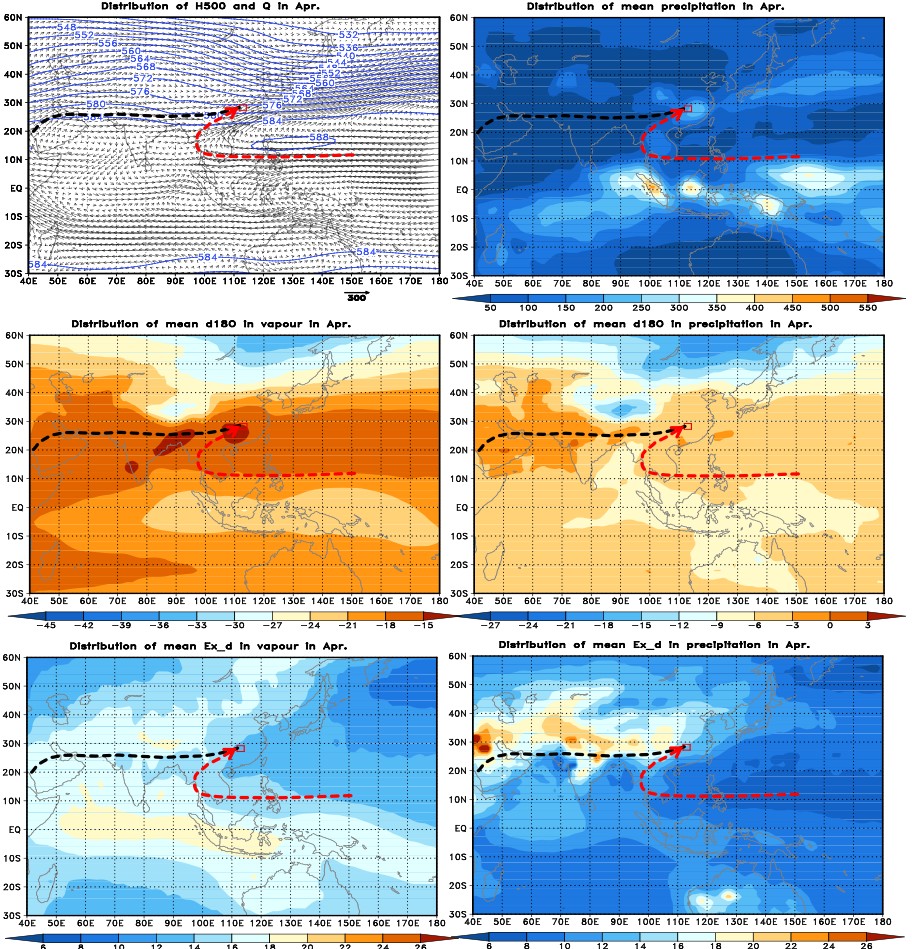

Fig. 5 Mean vapor transport paths to the Dongting Lake Basin and the spatial

distributions of $Q$ with $H_{500}$ (a), $P$ (b), $\delta^{18}O_v$ (c), $\delta^{18}O_p$ (d), Ex_d$_v$ (e), and Ex_d$_p$ (f) in

April.

Compared to the situations in January, there were no major changes in the spatial

distributions of $\delta^{18}O_v$ and $\delta^{18}O_p$ in April (Figs. 5c and 5d). In the regions with low $\delta^{18}O_v$

and $\delta^{18}O_p$ values, previously located in Eastern Siberia and the Tibetan Plateau,

atmospheric stable isotopes have significantly enriched. Due to temperature rise and

enhanced evaporation, the regions with high levels of $\delta^{18}O_v$ and $\delta^{18}O_p$ in the mid-to-





low latitudes showed continual increases in the $\delta^{18}O_v$ and $\delta^{18}O_p$ values. With the
strengthening of water vapor convergence in the Dongting Lake Basin, the $\delta^{18}O_v$ in the
Dongting Lake Basin showed significant increases in April, leading to an isotopic
enrichment in precipitation. Compared to the large-scale region, the water isotope
composition in the Dongting Lake Basin, was not significantly different from that of
the mid-to-low latitude ocean, indicating the controls by the maritime air masses (Figs.
5c and 5d).

In April, the spatial distributions of Ex_$d_v$ and Ex_$d_p$ were comparable to the

situations in January (Figs. 5e and 5f). The regions with high-value Ex_$d_v$, previously
located in Eastern Siberia and the Tibetan Plateau, respectively, showed significant
reductions in range and intensity, but the regions with low-value Ex_$d_v$ in the Western
Pacific expanded, thereby reducing the differences between land and sea. With the
continuous inland influx of maritime water vapor from the Western Pacific Ocean, the
range of low-value regions of the Ex_$d_p$ has expanded. Influencing by the increasing
precipitation, the range of high-value regions of the Ex_$d_p$ in mid-latitude inland
regions has narrowed, but the intensity has increased to varying degrees, especially in
West Asia. Finally, both the Ex_$d_v$ and Ex_$d_p$ in the Dongting Lake Basin showed
decreases, which were influenced by the gradually strengthening summer monsoon and
the situation of water vapor transport (Figs. 5e and 5f).

Based on the vector interpolation method, two water vapor transport paths were

obtained regarding the Changsha site as the endpoint (Fig. 5a). The first water vapor
transport path—that is, the Path I (represented by black arrow lines in Fig. 5), was



essentially consistent with the water vapor transport path in January, but it is slightly
shifted northward by one degree of latitude. The second water vapor transport path—
that is, Path II (represented by red arrow lines in Fig. 5), driven by the weak Western
Pacific subtropical high, guided warm and moist water vapor from the low latitudes of
the Western Pacific along the outer edge of the subtropical high, passing through the
South China Sea and the Indochinese Peninsula and finally reached into the Dongting
Lake Basin. Corresponding data at the grid points along two water vapor transport paths
were extracted and plotted for the $Q$, $P$, $\delta^{18}O_v$, $\delta^{18}O_p$, Ex_$d_v$, and Ex_$d_p$ (Fig. 6).

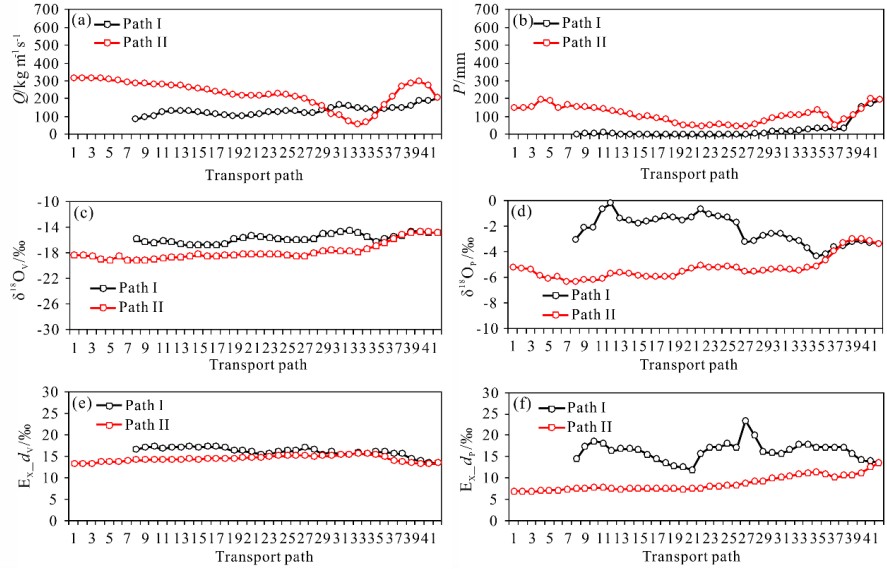


Fig. 6 Mean variations of $Q$ (a), $P$ (b), $\delta^{18}O_v$ (c), $\delta^{18}O_p$ (d), Ex_$d_v$ (e), and Ex_$d_p$ (f)
along the vapor transport paths in April.
Along Path I, both the $Q$ and $P$ showed slight increases compared to the situations
in January (Figs. 6a and 6b), with the average values in the first half of the water vapor
transport path before entering the Dongting Lake Basin were 128.23 kg m$^{-1}$ s$^{-1}$ and 10.5



mm, respectively, still at relatively low levels. After entering the Dongting Lake Basin,
the average $Q$ and $P$ increased to 180.13 kg m$^{-1}$ s$^{-1}$ and 135.0 mm, respectively. Under
the transport of latitudinal water vapor, the $\delta^{18}O_v$ increased slightly from −15.83‰ to
−14.85‰, while the $\delta^{18}O_p$ decreased slightly from −2.06‰ to −3.28‰ (Figs. 6c and
6d). The corresponding Ex_d$_v$ decreased from 16.45‰ to 14.26‰, and the Ex_d$_p$
decreased from 16.60‰ to 14.97‰, indicating the input of oceanic water vapor (Figs.
6e and 6f).

Along Path II, both the $Q$ and $P$ were significantly larger than those along the

latitudinal Path I (Figs. 6a and 6b), with the average values before entering the Dongting
Lake Basin were 226.6 kg m$^{-1}$ s$^{-1}$ and 108.2 mm, respectively. After entering the
Dongting Lake Basin, these values increased to 269.2 kg m$^{-1}$ s$^{-1}$ and 148.6 mm,
respectively. The area where $Q$ decreased significantly corresponds to a water vapor
divergence region at the southwest corner of the Indochinese Peninsula. Under the
meridional water vapor transport, the $\delta^{18}O_v$ increased from −18.21‰ to −14.86‰,
while the $\delta^{18}O_p$ from −5.50‰ to −3.15‰ (Figs. 6c and 6d); correspondingly, the Ex_d$_v$
decreased from 14.61‰ to 13.50‰, while the Ex_d$_p$ increased from 8.32‰ to 11.81‰
(Figs. 6e and 6f), following the variation rule of excess deuterium during water vapor
transport (Vasil'chuk, 2014).
**3.2.3 Average Water Vapor Transport Path in the Dongting Lake Basin in June**

Based on the ERA5 reanalysis data and isoGSM2 simulation data, the spatial

distributions of the average $H_{500}$, $Q$, $P$, $\delta^{18}O_v$, $\delta^{18}O_p$, Ex_d$_v$, and Ex_d$_p$ were
respectively calculated and plotted in June (Fig. 7). At the $H_{500}$ field (Fig. 7a), the East



Asian Trough continues to stably exist in June, but the position of trough line shifted
eastward over the North Pacific Ocean. The high-pressure ridge in the eastern part of
the Ural Mountains weakened and shifted eastward over Lake Baikal. The rapidly
intensifying western Pacific subtropical high expanded westward and northward, with
its ridge line located at approximately 22~23°N, while the India-Burma Trough in the
northern Bay of Bengal strengthened continuously. The atmospheric circulation
situation indicated that most of East Asia, including the south of the Yangtze River, has
entered the prevailing period of summer monsoon (Fig. 7a). The warm and moist water
vapor from the Arabian Sea and the Bay of Bengal driven by the India-Burma Trough
as well as along the outer edge of the subtropical high from the western Pacific met the
cold air moving southward behind the East Asian Trough and thus generated an
extremely long rain belt spanning 20 degrees of latitude and 70 degrees of longitude
from India, through the Indochinese Peninsula, to southern China until central Japan
(Fig. 7b). In this rain belt, the three largest precipitation centers were located on the
west coast of India, the Thai-Myanmar border region, and the Jiangnan region of China.
The formation of the first two precipitation centers was related to terrain, while the
formation of the precipitation center in the Jiangnan region of China was related to the
convergence of warm and moist water vapor from low latitudes to this region (Fig. 7b).

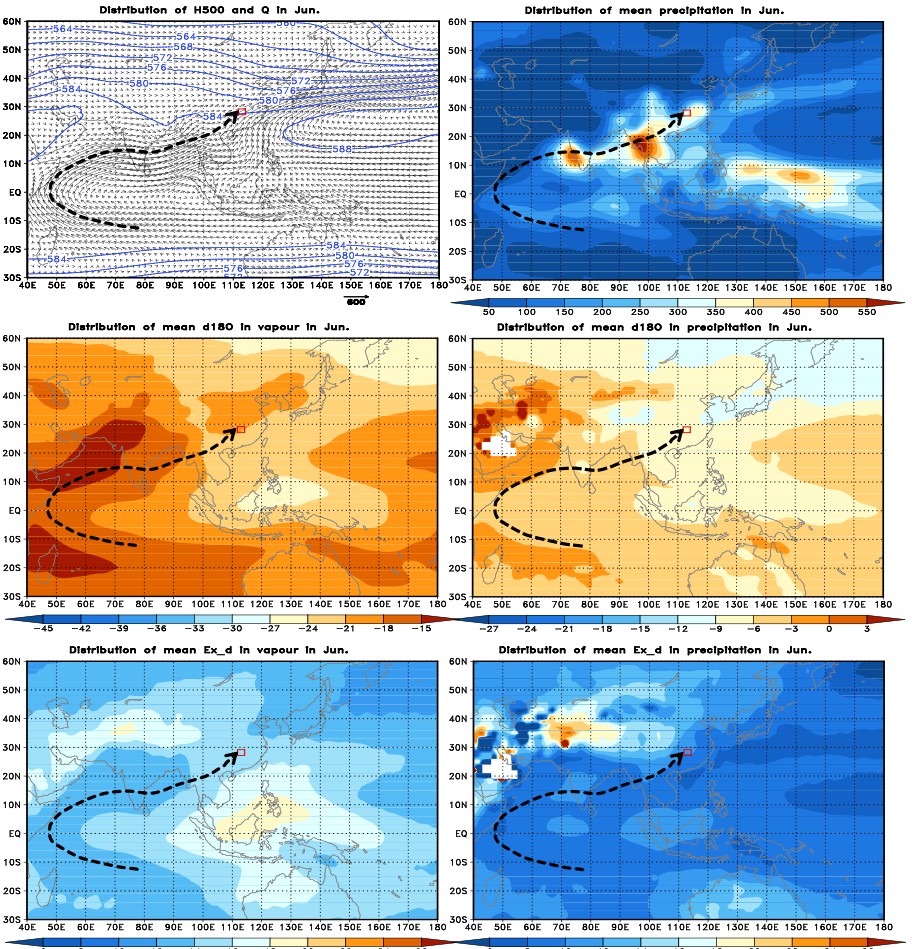


Fig. 7 Mean vapor transport path to the Dongting Lake Basin and the spatial
distributions of $Q$ with $H_{500}$ (a), $P$ (b), $\delta^{18}O_v$ (c), $\delta^{18}O_p$ (d), Ex_$d_v$ (e), and Ex_$d_p$ (f) in
June.
With the change in circulation situation, the distributions of the $\delta^{18}O_v$ and $\delta^{18}O_p$ in
June changed accordingly (Figs. 7c and 7d). The stable isotopes in both water vapor
and precipitation at mid-high latitudes were significantly enriched, with reduced spatial
differences. The $\delta^{18}O_v$ and $\delta^{18}O_p$ values remained high in the Arabian Sea, the Bay of
Bengal, and the Southern Ocean, the region with high $\delta^{18}O$ in the western Pacific



became narrowed, and the low $\delta^{18}O$ values in the Indonesia-Philippines region in the
western equatorial Pacific were associated with the enhanced water vapor convergence
(Figs. 7a, 7c, and 7d). The regional low $\delta^{18}O$ center previously present in the cold
season over the Tibetan Plateau had disappeared. In the Jiangnan region of China, the
convergence of water vapor from low-latitude oceans led to an isotopic enrichment in
water vapor, but the strong rainout effects caused an isotopic depletion in precipitation
(Figs. 7c and 7d). The spatial distributions of Ex_$d_v$ and Ex_$d_p$ in June showed no
significant differences compared to the situations in April (Figs. 5e, 5f, 7e, and 7f). The
high-value regions for the Ex_$d_v$ and Ex_$d_p$ were located in the region stretching from
western Asia through the Tibetan Plateau to southwestern China, as well as in the vast
oceanic region centered around the Philippines and Indonesia. Under the influence of
the summer monsoon, the difference in deuterium excess between East Asia and its
water vapor source—that is, the low-latitude ocean, remained relatively small (Figs. 7e
and 7f).
Based on the vector interpolation of the $Q$ field (Fig. 7a), the water vapor transport
path regarding the Dongting Lake Basin as the endpoint was determined in June (shown
by the black arrow lines in Fig. 7). This water vapor transport path originated from the
northern branch of the South Indian Ocean subtropical high, crossed the equator, and
transported through the Somali Sea, the Arabian Sea, the Indian Peninsula, the Bay of
Bengal, the Indochinese Peninsula, and entered the southwestern region of China,
finally reaching the Dongting Lake Basin. It can be seen that this water vapor transport
path was consistent with the prevailing wind direction in June as shown in Fig. 1a. The





corresponding $Q$, $P$, $\delta^{18}O_v$, $\delta^{18}O_p$, Ex_d$_v$, and Ex_d$_p$ along the water vapor transport
path were extracted, and plotted in Fig. 8.

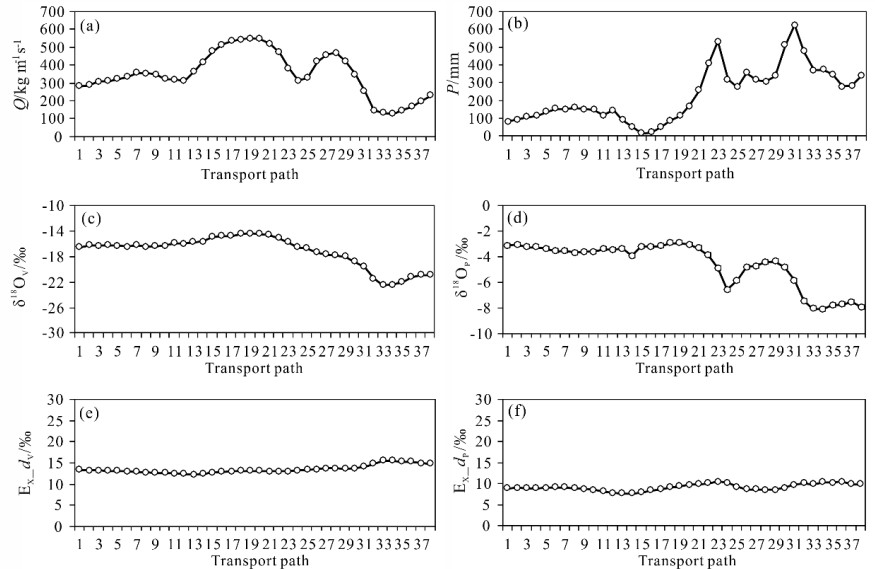


Fig. 8 Mean variations of $Q$ (a), $P$ (b), $\delta^{18}O_v$ (c), $\delta^{18}O_p$ (d), Ex_d$_v$ (e), and Ex_d$_p$ (f)

along the vapor transport path in June.

Along the water vapor transport path, the average $Q$ and $P$ were at their maximum

throughout the year, reaching 353.86 kg m$^{-1}$ s$^{-1}$ and 236.5 mm, respectively (Figs. 8a
and 8b). The three extreme values of the $Q$ along the transport path, or in the process
of transitioning from the maximum to minimum values, correspond to three $P$ extremes
located at the western coast of the Indian Peninsula, the border region between Thailand
and Myanmar, and the region around the Dongting Lake Basin (Fig. 7b), with the values
of the three $P$ extremes of 535.1 mm, 627.8 mm, and 341.5 mm, respectively (Fig. 8b).
With continuous precipitation especially after experiencing heavy precipitation and the
simultaneous persistent rainout processes, the stable isotopes in both water vapor and



precipitation exhibit a trend of continuous depletion (Figs. 8c and 8d). However, due to

continuous water vapor supply from low-latitude oceans, there were no significant

changes in both the Ex_$d_v$ and Ex_$d_P$ (Figs. 8e and 8f).

**3.2.4 Average Water Vapor Transport Path in the Dongting Lake Basin in October**

The spatial distributions of the average $H_{500}$, $Q$, $P$, $\delta^{18}Ov$, $\delta^{18}O_P$, Ex_$d_v$, and Ex_$d_P$

in October were shown in Fig. 9. A notable feature at the $H_{500}$ field in October was the

expansion of the latitudinal westerlies toward lower latitudes (Fig. 9a). In East Asia,

westerly winds prevail in the inland regions north of approximately 30°N, while much

of the regions south of 30°N were still influenced by the subtropical high-pressure

system. Compared to the peak period, the West Pacific subtropical high had

significantly weakened in autumn, and its main body had also retreated to the open sea.

However, a mesoscale anticyclone split from the high still controlled the Jiangnan

region of China including the Dongting Lake Basin, creating a climate characterized by

clear and crisp autumn (Fig. 9a). Due to the disappearance of the India-Burma Trough

and influenced by the anticyclone circulation, the water vapor transport from the

southwest low-latitude oceans decreased significantly. In the Dongting Lake Basin,

both the meridional and latitudinal water vapor transport were even less than the values

in January (Fig. 3a and 9a; Xiao et al., submitted). Apart from the autumn rains in

western China, precipitation was generally scarce in East Asia in this period, with the

rain belt shifting southward to lower latitudes corresponding to the convergence zone

near the equator, with the largest precipitation regions located respectively south of the

Equator in the Indian Ocean, the Malay Peninsula, and north of the Equator in the



western Pacific (Fig. 9b).

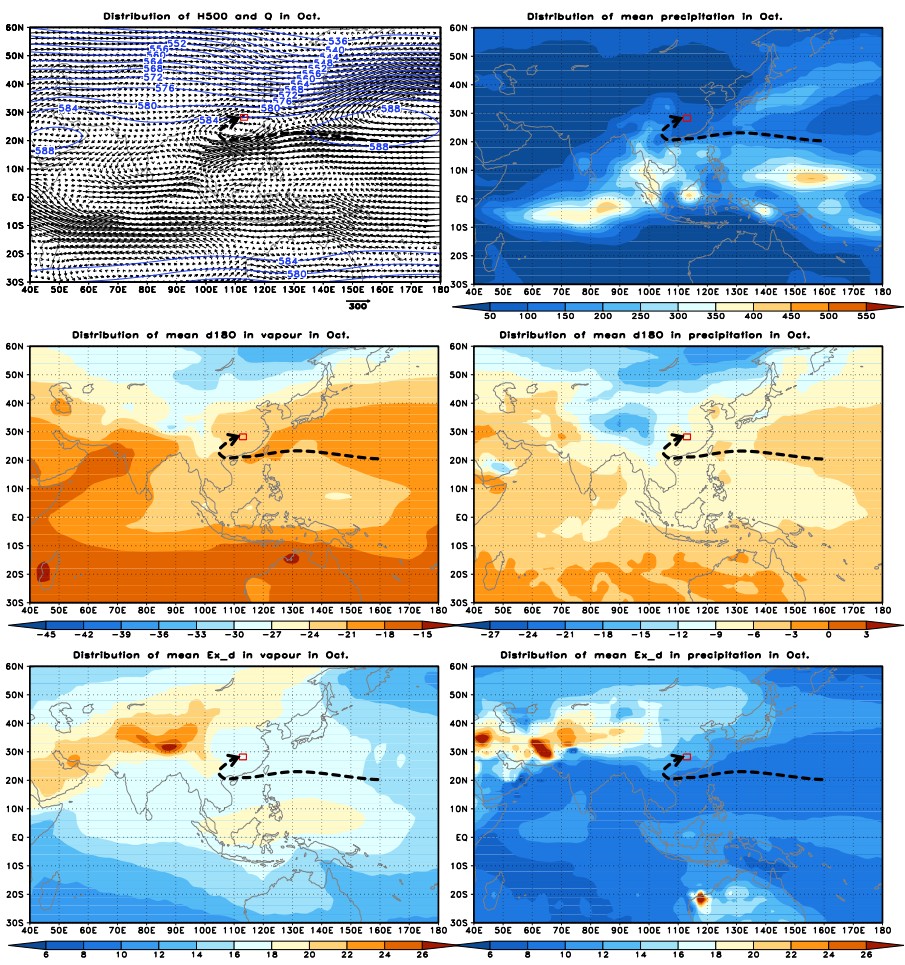


Fig. 9 Mean vapor transport path to the Dongting Lake Basin and the spatial

distributions of $Q$ with $H_{500}$ (a), $P$ (b), $\delta^{18}O_v$ (c), $\delta^{18}O_p$ (d), Ex_$d_v$ (e), and Ex_$d_p$ (f) in

October.

Compared to the situations in June, there were no significant changes in the spatial

distribution of the $\delta^{18}O_v$ and $\delta^{18}O_p$ in October, but their differences between land and
sea as well as between high and low latitudes increased largely (Figs. 9c and 9d). The
stable isotopes in water vapor and precipitation were significantly depleted in Eastern



587 Siberia and the Tibetan Plateau, and accompanied by expansion of ranges. As a result,

588 the $\delta^{18}O_v$ and $\delta^{18}O_p$ showed a significant decrease in the inland regions north of

589 approximately 30°N, but unchanging in most regions south of 30°N. From the spatial

590 distributions in October, both the Ex_$d_v$ and Ex_$d_p$ over the ocean or on land showed

591 increases with varying degrees (Figs. 9e and 9f). The high-value regions of the Ex_$d_v$

592 were distributed along a line from the Arabian Peninsula, West Asia, the Tibetan Plateau,

593 to Eastern Siberia, with the maximum value located in the Tibetan Plateau. The high-

594 value region of Ex_$d_p$ was distributed from the Arabian Peninsula, West Asia, the

595 Tibetan Plateau, to the Yunnan-Guizhou Plateau. With the weakening of the summer

596 monsoon, the Ex_$d_v$ and Ex_$d_p$ were not significantly different in East Asia including

597 the Dongting Lake Basin from those in the surrounding oceans.

598  Based on the vector interpolation of the $Q$ field (Fig. 9a), the water vapor transport

599 path regarding the Changsha site as the endpoint was determined in October (indicated

600 by the black arrow line in Fig. 9). This water vapor transport path originated from the

601 western Pacific, passed through the South China Sea, flowed westward along the

602 easterly jet located in the south of the West Pacific Subtropical High and of the

603 anticyclonic circulation over the Jiangnan region in China, and finally entered the

604 Dongting Lake Basin bypassing the southwest of the anticyclone. Although this water

605 vapor path belonged to the latitudinal transport, the water vapor source originated from

606 the low-latitude oceans of the western Pacific.

607  The corresponding $Q$, $P$, $\delta^{18}O_p$, $\delta^{18}O_v$, Ex_$d_v$, and Ex_$d_p$ along the water vapor

608 transport path were derived in October (Fig. 10). Under the stable atmospheric





circulation conditions, the average water vapor flux steadily decreased from
approximately 299.0 kg m$^{-1}$ s$^{-1}$ in the source region to 168.0 kg m$^{-1}$ s$^{-1}$ along the water
vapor transport path, and further below 100.0 kg m$^{-1}$ s$^{-1}$ after entering the Dongting Lake
Basin (Fig. 10a). The $P$ values changed gradually along the water vapor transport path,
decreasing from the initial approximately 145 mm to below 100 mm in the Dongting
Lake Basin, and further to 60.5 mm in the Changsha region (Fig. 10b). Both the $\delta^{18}O_v$
and $\delta^{18}O_p$ values showed slow decreases as the ranges within 2.0‰ and 1.0‰
respectively (Figs. 10c and 10d). The Ex_$d_v$ showed minor fluctuation, stabilizing at
approximately 16.8‰, while the corresponding Ex_$d_p$ remained around 8.62‰ before
entering the Dongting Lake Basin. Due to the replenishment from the surface
evaporation, the Ex_$d_p$ values increased to 11.60‰ after entering the Dongting Lake
Basin (Figs. 10e and 10f).

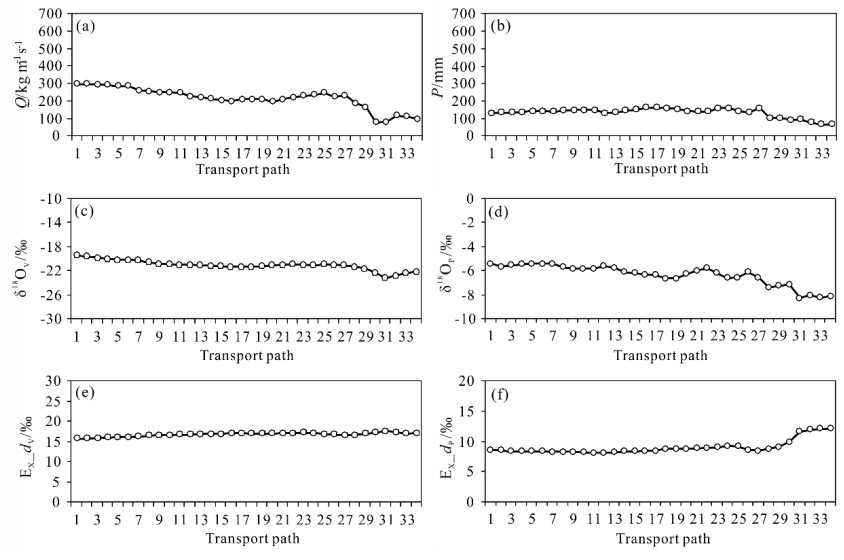


Fig. 10 Mean variations of $Q$ (a), $P$ (b), $\delta^{18}O_v$ (c), $\delta^{18}O_p$ (d), Ex_$d_v$ (e), and Ex_$d_p$ (f)
along the vapor transport path in October



**4. Discussion**

**4.1 The Influences of the Seasonality in Water Vapor Sources on the Precipitation Isotopes.**

The comparisons between the $Q$ and $\delta^{18}O_p$ in the representative months indicated that there seems to be no obvious correspondence between these two factors: the months with low $Q$ would exhibit either high or low $\delta^{18}O_p$, e.g. January and October, respectively (Figs. 3 and 9). Similarly, the months with high $Q$ would exhibit either low or high $\delta^{18}O_p$, for example, June and April, respectively (Figs. 5 and 7). It has been found that regardless of the season, the precipitation in the Dongting Lake Basin mainly originated from warm and moist water vapor in low latitudes (Figs. 3, 5, 7, and 9). Therefore, whether the water vapor isotopes at the source regions and along the transport path influence the downstream isotopes of precipitation or water vapor? To reveal this causality, after considering the water vapor transport paths and the air mass properties of water vapor in the representative months, the regions corresponding to the Arabian Peninsula (40°E~56°E, 16°N~28°N), the Arabian Sea (56°E~74°E, 10°N~20°N), the Bay of Bengal (80°E~98°E, 8°N~18°N), the western Pacific Ocean (120°E~160°E, 6°N~20°N), the Dongting Lake Basin (110°E~114°E, 25°N~30°N), and the inland regions of East Asia monsoon region (110°E~135°E, 42°N~55°N) were labeled as Regions I, II, III, IV, V, and VI, respectively (Fig. 11). The average $\delta^{18}O$ and Ex_d of water vapor and precipitation for each representative region were calculated in January, April, June, and October, respectively. Since the seasonal variations in the $\delta^{18}O_v$ were similar to that in $\delta^{18}O_p$, Table 1 only provided the average $\delta^{18}O$ and Ex_d





of water vapor for each representative region.

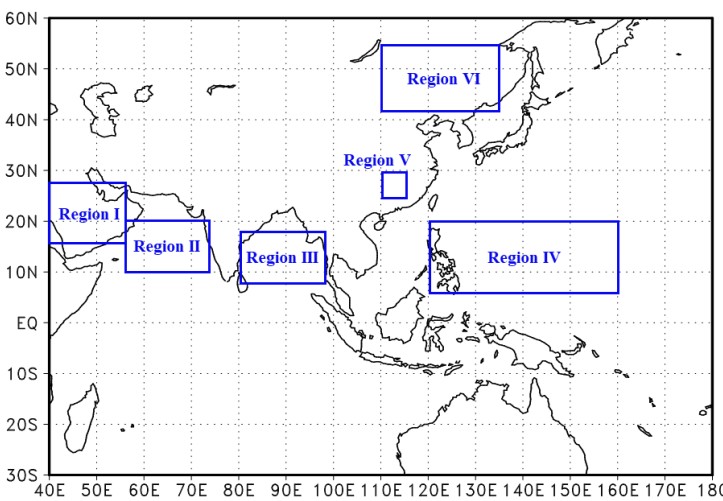

Fig. 11 Geographical distribution of representative regions

(Region I: Arabian Peninsula, Region II: Arabian Sea, Region III: Bay of Bengal,

Region IV: Western Pacific, Region V: Dongting Lake Basin, Region VI: Inland of the

East Asian monsoon region at middle and high latitudes)

Not only in Regions I to V located at mid to low latitudes but also in Region VI

located in the mid to high latitude inland regions, there were significant seasonal
variations in the average $\delta^{18}O_v$ and Ex_$d_v$ (Table 1). The seasonal differences in the
$\delta^{18}O_v$ (the differences between the monthly maximum and minimum values) in these
six representative regions were 2.94‰, 3.34‰, 4.19‰, 5.06‰, 7.18‰, and 18.94‰,
respectively, with the largest seasonal difference in $\delta^{18}O_v$ appeared in Region VI (Table
1). Except for Region VI, the minimum values of the monthly $\delta^{18}O_v$ in other
representative regions, all occurred in October, while the maximum or second
maximum values occurred in April. The seasonal differences in the Ex_$d_v$ in these six



representative regions were 4.69‰, 5.42‰, 3.56‰, 3.81‰, 3.59‰, and 9.31‰,
respectively, with the largest seasonal difference in the Ex_$d_v$ still in Region VI. Except
for Region VI, the maximum values of monthly Ex_$d_v$ in other representative regions
mostly occurred in October, while the minimum or second minimum values occurred
in April or June (Table 1). These results indicated significant differences in water vapor
isotopes between Region VI and other representative regions.
Table 1 Mean $\delta^{18}O_v$ and Ex_$d_v$ for representative regions in the representative months

| Factors | Months | Region I | Region II | Region III | Region IV | Region V | Region VI |
|---|---|---|---|---|---|---|---|
| $\delta^{18}O_v$ | January | -19.04 | -17.42 | -18.72 | -17.24 | -18.82 | -40.93 |
|  | April | -16.22 | -15.78 | -17.83 | -17.94 | -14.91 | -28.70 |
| /‰ | June | -17.10 | -14.45 | -18.10 | -21.92 | -20.77 | -21.99 |
|  | October | -19.16 | -17.79 | -22.02 | -22.30 | -22.09 | -29.32 |
| Ex_$d_v$ | January | 18.45 | 18.51 | 17.22 | 14.39 | 15.21 | 23.20 |
|  | April | 16.93 | 17.46 | 17.03 | 13.91 | 13.55 | 15.64 |
| /‰ | June | 16.04 | 13.09 | 13.98 | 16.25 | 15.03 | 13.89 |
|  | October | 20.73 | 17.33 | 17.54 | 17.72 | 17.14 | 18.29 |

According to the statistics in Table 1, in January, the average $\delta^{18}O_v$ and Ex_$d_v$ were
−19.04‰ and 18.45‰, respectively, in Region I under the latitudinal water vapor
transport, while −18.82‰ and 15.21‰, respectively, in Region V with their differences
only 0.22‰ and 3.24‰, respectively; In April, the average $\delta^{18}O_v$ and Ex_$d_v$ were
−16.22‰ and 16.93‰, respectively, in Region I also under the latitudinal water vapor
transport, while −17.94‰ and 13.91‰, respectively, in Region IV under the meridional
water vapor transport, and −14.91‰ and 13.55‰, respectively, in Region V; In June,
the average $\delta^{18}O_v$ were −14.45‰ and −18.10‰, respectively, the average Ex_$d_v$ values



were 13.09‰ and 13.98‰, respectively, in Regions II and III, all under the meridional
water vapor transport, while the average $\delta^{18}O_v$ and average Ex_$d_v$ were −20.77‰ and
15.03‰, respectively, in Region V after experiencing intense rainout processes; In
October, the average $\delta^{18}O_v$ and Ex_$d_v$ were −22.30‰ and 17.72‰, respectively, in
Region IV under the weakened meridional water vapor transport, showing non-
significantly differences from the values of −22.09‰ and 17.14‰, respectively, in
Region V (Table 1).

Furtherly, by comparing the water vapor isotopes in Region V with those in Region

VI, it can be found that although both regions were all located in the East Asian
monsoon region, there were differences in the seasonal variations of water vapor
isotopes (Table 1; Fig. 11). For instance, the average $\delta^{18}O_v$ in Regions V and VI in all
of the representative months were −19.15‰ and −30.24‰, respectively, with a
difference of 11.09‰. Moreover, the average $\delta^{18}O_v$ of these two regions showed the
largest differences with a value of 22.11‰ in January, which represented the peak of
the winter monsoon, while in June which represented the peak of the summer monsoon,
the difference was only 1.22‰. The water vapor isotopes in Region V were consistently
enriched compared to those in Region VI. The average Ex_$d_v$ in Regions V and VI in
all of the representative months were 15.23‰ and 17.76‰, respectively, with a
difference of −2.52‰, which was not too large. The difference was largest in January,
reaching −7.99‰, while in June, the difference was only 1.14‰, indicating that the
water vapor sources during the summer monsoon were similar in these two regions
(Table 1).




The above results about the water stable isotope differences between the
representative regions conform to the latitudinal and continental effects of water stable
isotopes and follow the law of material migration, which states that the composition of
water stable isotopes becomes more depleted with increasing latitude and water vapor
transport from high to low value regions (Feng et al., 2009; Zhang et al., 2012; Zhang
et al., 2016). With emphasis, for the water vapor source of precipitation in the Dongting
Lake Basin, the oceanic representative regions located at low latitudes may not
necessarily be the initial water vapor source regions, and the relationship between
upstream and downstream regions may not entirely be point-to-point, as there were
continuous water recycling and rainout processes along the water vapor transport path
(Pokam et al., 2012; Risi et al., 2013; Christner et al., 2018). However, through the
comparisons above, it can be observed that the influences of upstream regions on the
water vapor amount and water vapor isotopes in downstream regions during water
vapor transport were significant.
**4.2 Isotopic Properties of Air Masses**
According to the definition of meteorology, air mass refers to a large-scale body
of air over land or sea with relatively uniform horizontal physical properties such as
temperature, humidity, and atmospheric stability. The horizontal extent of an air mass
ranges from $10^2$ km to $10^3$ km, and the vertical extent ranges from $10^0$ km to $10^1$ km,
while within the same air mass, there is little variation in temperature gradients,
atmospheric vertical stability, and weather phenomena (Zhou et al., 1997). Under large-
scale and relatively uniform underlying surfaces and stable atmospheric circulation



conditions, water vapor and its transport belong to the characteristics of air masses or
have the properties of the air mass origin regions (Dettinger, 2013; Lavers et al., 2013).
Considering the sources and sinks of water vapor, the spatial distribution of water vapor
isotopes is relatively similar within an air mass. In maritime air masses, water vapor
isotopes are relatively enriched, while deuterium excess of water vapor is relatively
more negative, while in continental air masses, water vapor isotopes are relatively
depleted, while excess deuterium of water vapor is relatively more negative (Rozanski
et al., 1993; Araguás-Araguás et al., 1998).
With the seasonal variation in the position of the sun's orbit, the atmospheric
circulation conditions undergo seasonal variations and thus lead to the seasonality of
the air masses properties (Qian et al., 2009; Parding et al., 2016). The abundance of
water vapor isotopes at a fixed location varies due to variations in circulation conditions
(Lacour et al., 2018; Dee et al., 2018; Gou et al., 2022). In this study, the isotopic
compositions of water vapor in maritime Regions II and IV at low latitudes and in
inland Region VI at high latitudes exhibited significant seasonal variations due to
interactions between tropical continental air masses (located in southern West Asia) and
tropical maritime air masses, between tropical maritime air masses and equatorial air
masses, and between temperate continental air masses and temperate maritime air
masses, respectively (Table 1; Fig. 11). In the process of seasonal changes, as air masses
move out of their source regions, their physical and weather characteristics also change
with the variations in underlying surface properties and large-scale vertical motion
conditions. East Asia is primarily controlled by modificatory air masses (Ding, 1990;



Chang et al., 2012). Whether cold and dry air masses moving southward or warm and
moist air masses moving northward, the isotopic composition of water vapor in the
modificatory air mass continues to become more negative, while the deuterium excess
of water vapor continues to become more positive than the original air mass (Zhou et
al., 2019; Xu et al., 2019; Jackisch et al., 2022). In this study, interactions between
modificatory marine air mass and modificatory continental air mass result in the water
vapor isotope in Region V that differed from maritime air masses and continental air
masses (Table 1; Fig. 11). In summary, as an important member of the climate system,
air mass possesses not only thermodynamic, dynamic, hydrous and static properties but
also isotopic properties.
**4.3 The Difference Between Water Vapor Field and Wind Field**
The water vapor flux $Q$ reflects the direction and magnitude of water vapor
transport in the atmosphere, while wind $V$ reflects the direction and magnitude of the
movement of air particles in the atmosphere (Feng et al., 2009; Zhang et al., 2012;
Zhang et al., 2016). There are both differences and connections between the two factors.
Water vapor is transported by wind, and the wind carries water vapor from one place to
another, and the directions of water vapor and wind may be consistent, inconsistent, or
even opposite. In the East Asian monsoon region, the prevailing wind direction during
the summer monsoon period is generally consistent with the average water vapor
transport direction, both being southwest or southeast direction (Barker, et al., 2015;
Wu et al., 2015; Tang et al., 2015). In this study, the water vapor transport path was
consistent with the prevailing wind direction in June (Figs. 1a and 9). However, during

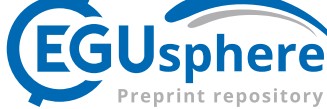

the winter monsoon period, the prevailing wind direction may not be consistent with
the average transport direction of water vapor—that is, the prevailing wind direction in
January was northwest or northeast direction, while the average transport direction of
water vapor in this period was southwest or southeast direction (Figs. 1a and 3), and
supported by the water vapor transport study focusing on the Changsha region (Xiao et
al., submitted).

Previous studies have shown that the most common weather systems and most

precipitation events in the East Asian monsoon region are caused by cold fronts
resulting from the interaction of warm and cold air masses (Chen et al., 2020).
According to classical meteorological theory (Zhou et al., 1997), in a cold front system,
there appears the wind from the southwest direction blows ahead of the front, and a
northwest wind blows behind the front as shown in the schematic diagram in Fig. 12a.
Warm and moist air from low latitudes lifts along the front and leads to rainfall, while
cold and dry air from high latitudes moves southward beneath the front and lifts the
warm and moist air. At different altitudes and positions, the directions of air particle
movement and water vapor transport are different. For example, at the point A located
above the warm and moist air side of the cold front surface, both air particles and water
vapor are transported by southwest wind. At the point C located below the cold, dry air
side of the cold front surface, both air particles and water vapor are transported by
northwest wind. However, at the point B located within the front zone, the wind
direction and speed are uncertain (Fig. 12b). Therefore, the dominant wind directions
may not always align with the average water vapor transport direction, especially in





frontal weather systems that dominate precipitation in the East Asian monsoon region.

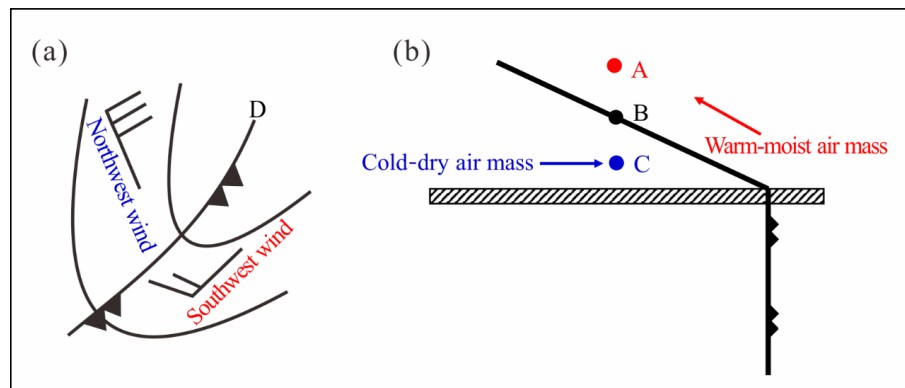

Fig. 12 Schematic diagram of a cold front system in East Asia (based on Zhou et al.,

1997).

**4. Conclusion**
Our findings revealed significant influences of water vapor source and transport
on precipitation isotopes in Dongting Lake Basin. Specifically, in January, water vapor
contributing to the Dongting precipitation originated near the Arabian Peninsula and
was driven by the southern branch of the westerly stream jet on the southern side of the
Tibetan Plateau, water vapor transported along the southern side of the Tibetan Plateau,
passed through southwestern China via the northern part of the Indian Peninsula, and
reached the Dongting Lake Basin. In April, two distinct water vapor transport paths
contributed to the Dongting precipitation, the first followed a trajectory similar to the
average water vapor transport path in January, albeit shifted slightly northward by one
degree of latitude. The second transport path, driven by the weak Western Pacific
subtropical high, guided warm and moist water vapor from the low latitudes of the
Western Pacific along the outer edge of the subtropical high, passing through the South



China Sea and the Indochinese Peninsula and finally reached into the Dongting Lake
Basin. In June, the Dongting precipitation was influenced by a water vapor transport
path originating from the northern branch of the South Indian Ocean subtropical high,
crossed the equator, and transported through the Somali Sea, the Arabian Sea, the Indian
Peninsula, the Bay of Bengal, the Indochinese Peninsula, and entered the southwestern
region of China, finally reaching the Dongting Lake Basin. In October, the average
water vapor transport path originated from the western Pacific, passed through the
South China Sea, flowed westward along the easterly jet located in the south of the
West Pacific Subtropical High and of the anticyclonic circulation over the Jiangnan
region in China, and finally entered the Dongting Lake Basin bypassing the southwest
of the anticyclone. Although this water vapor path belonged to the latitudinal transport,
the water vapor source originated from the low-latitude oceans of the western Pacific.
In these four months that representing different seasons, variations in the $\delta^{18}O$ and Ex_d
of precipitation and water vapor along these water vapor transport paths adhered to
principles of Rayleigh fractionation and water balance principles, underscoring the
complex transport paths and processes that influence isotopic variations in precipitation
in the Dongting Lake Basin. However, the prevailing wind direction may not be
consistent with the average transport direction of water vapor, especially during the
winter monsoon, which can be explained by the water vapor field and wind field in
frontal weather systems that dominate precipitation in the East Asian monsoon region.

Overall, the approach utilized in this study is grounded in fundamental

meteorological theories, specifically involving water vapor diagnosis and calculations



(including source regions, transport paths, and transport quantities), this analytical
method is robust and has a clear physical basis. The scientific question addressed by
this study is not centered on the spatial and temporal variations of water isotopes, but
rather on how the seasonal variations in regional precipitation and precipitation isotopes
respond to the seasonal variations in water vapor sources and transport.
**Competing interests**
The authors declare that they have no known competing financial interests or personal
relationships that could have appeared to influence the work reported in this paper.
**Acknowledgments**
This study was supported by the Natural Science Foundation of Hunan Province, China
(No. 2023JJ40445) the National Natural Science Foundation of China (No. 42101130),
and the Aid Program for Science and Technology Innovative Research Team in Higher
Educational Institutions of Hunan Province (0531120-4944). We are grateful to the
graduate students who laboriously sampled water samples without interruption and
tested water stable isotopes for 13 years.
**Author contribution**
Xiong Xiao: Methodology, Software, Writing- Original draft preparation, Reviewing
and Editing. Xinping Zhang: Supervisor, Guide, Data curation, Methodology, Software,
Writing- Original draft preparation, Reviewing and Editing. Zhuoyong Xiao:
Methodology, Writing-Original draft preparation. Zhongli Liu, Dizhou Wang, Cicheng
Zhang, Zhiguo Rao, Xinguang He, and Huade Guan: Methodology, Reviewing and
Editing. All authors made substantial contributions to the discussion of content.



**Code/Data availability**

The global atmospheric reanalysis data and water stable isotope simulation data are downloaded from the ECMWF 5th generation atmospheric reanalysis data (ERA5, https://cds.climate.copernicus.eu/) and the second-generation isoGSM2 dataset (https://datadryad.org/stash/dataset/doi:10.6078/D1MM6B), respectively. The stable isotopic data of precipitation and meteorological data at the Changsha station are accessible by emailing the corresponding author (zxp@hunnu.edu.cn) with a reasonable request.

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
