# Peer review of "Water Vapor Transport and its Influence on Water Stable Isotope in"

_EGUsphere, 2024_

## Author Comment (AC1)

*A detailed, point-by-point response to the review comments is given below. Each review comment is repeated followed with our action to modify the manuscript. All Page and Line numbers correspond to locations in the revised manuscript.*

Comments from Reviewer Luke He:

The manuscript entitled "Water Vapor Transport and its Influence on Water Stable Isotope in Dongting Lake Basin" by Xiao et al. presents a comprehensive study on the sources and transport pathways of water vapor and their influence on the isotopic composition of precipitation in the Dongting Lake Basin. This region provides an excellent case study for understanding the intricate links between atmospheric circulation, water vapor dynamics, and isotopic signatures in precipitation. The research is particularly significant in the context of climate change, where such insights can aid in predicting shifts in precipitation patterns and their isotopic composition, which are vital for water resource management and paleoclimatic interpretations. The authors have employed a multi-faceted approach, utilizing both observational data and model simulations, to trace the origins and pathways of water vapor contributing to precipitation in the study area. The use of reanalysis data and isotopic simulations adds depth to the analysis, allowing for a robust examination of the seasonal variations in water vapor transport and their isotopic implications. Overall, this manuscript could make a nice contribution and be of interest to many different groups ranging from hydrologists to meteorologists, and could be acceptablefor publication in the ACP. However, to further strengthen the manuscript and enhance its impact, several areas require the authors' attention.

Response: We appreciate the positive comments from the reviewer and have revised the manuscript accordingly, the details can be found in the responses to the specific concerns.

My major comments are:

Line 33-42: the Significance Statement, I suggest the following logical structure for the presentation: Starting from the sources of precipitation water vapor (which characterize the influence of atmospheric circulation), that is, the origins of water vapor (initial conditions), to the changes in water stable isotopes along the water vapor transport pathways (stable isotope fractionation and water vapor exchange), and finally to the changes in precipitation isotopes at the point of deposition (outcomes). This approach will reflect a comprehensive understanding of the cycle and fractionation of water stable isotopes within the context of atmospheric circulation.

Response: We appreciate the reviewer's constructive comments and have revised the structure of the Significance Statement: "This research explored how water vapor transports influenced the precipitation isotopes in the Dongting Lake Basin in representative months of different seasons. By tracking water vapor from its source regions, we revealed the influence of large-scale atmospheric circulation on the transportation of water vapor to the Dongting Lake Basin. The changes in water stable isotopes along the water vapor transport paths highlighted the isotopic fractionation and water vapor exchange that occurred along these paths, while the isotopic changes in the precipitation reflect the cumulative influences of water vapor transport on the

local precipitation. These comprehensive insights have clarified the influences of atmospheric circulation on water vapor transport and precipitation isotopes, and thus essential for predicting regional precipitation patterns" (Line 33-42).

Line 385-406 and the relevant descriptions: How are the water vapor pathways determined? How are the source regions of water vapor identified? And how are the scatter points along the water vapor transport pathways established?

Response: We appreciate the reviewer's constructive comments. For the comment "How are the water vapor pathways determined?", we added the relevant description in the manuscript as "The water vapor transport path was determined by the the rules to find the systematic vapor currents in the $Q$ field, which need to have the same directionality and draw the path along the central axis of the vapor currents" (Line 388-390); For the comment "How are the source regions of water vapor identified?", we added the relevant description in the manuscript as "The source regions of water vapor was determined based on the conditions for the formation of air masses, which need to have uniformity in the properties of the air mass and isotopes, and have a stable circulation field" (Line 390-393); For the comment "how are the scatter points along the water vapor transport pathways established?", we added the relevant description in the manuscript as "Moreover, the grid points along the water vapor transport path were identified on the central axis of the path and based on the principle of uniform distribution of the scatter points, and the factors at the grid points were obtained from these the scatter points" (Line 402-405).

Line 451~467: In April, there are two distinct pathways for water vapor transport. One is predominantly the transport of continental water vapor, and the other is maritime water vapor. A detailed comparison of the characteristic elements of these two pathways should be conducted. The water vapors from these two paths converge in the Dongting Lake Basin; which of these has a relatively more significant impaction the isotopic composition of precipitation in the Dongting Lake Basin region?

Response: We appreciate the reviewer's constructive comments. In April, we evaluate the impact of two water vapor transport paths on the isotopic composition of precipitation in the Dongting Lake Basin, considering the path with the air mass isotopic composition most similar to the precipitation isotopes in the basin, before its entry, as having the greater influence. According to this principle, we found that the input of oceanic air parcel with low deuterium excess had a stronger impact on the isotopic composition of precipitation in the Dongting Lake Basin. Referring to Table 1, we observed that in April, the $\delta^{18}O_v$ and $Ex\_d_v$ values of water vapor in the Dongting Lake Basin (i.e. Region V) were $-14.9$‰ and $13.6$‰, respectively. Meanwhile, before entering the Dongting Lake Basin, the $\delta^{18}O_v$ and $Ex\_d_v$ values of air parcel on the first water vapor transport path—that is, Path I, were $-14.9$‰ and $14.3$‰, respectively. For the second transport path—that is, Path II, these values were $-14.9$‰ and $13.5$‰, respectively. We can found that the oceanic air mass with low deuterium excess had a relatively more significant impact on the isotopic composition of precipitation in the

Dongting Lake Basin region. Therefore, we demonstrated these findings in the manuscript "For instance, two distinct water vapor transport paths were identified in April (Fig. 5), thus it is crucial to assess which path exerted a more significant influence on the isotopic composition of precipitation in the Dongting Lake Basin, with priority given to the path whose air parcel isotopic signature closely matches the precipitation isotopes in the basin before entry. According to this principle and referring to Table 1, we observed that in April, the average $\delta^{18}O_v$ and Ex_$d_v$ values of the Dongting Lake Basin (i.e. the Region V) were −14.9‰ and 13.6‰, respectively. Moreover, before entering the Dongting Lake Basin, the $\delta^{18}O_v$ and Ex_$d_v$ values of air parcel on the first water vapor transport path—that is, Path I, were −14.9‰ and 14.3‰, respectively (Figs. 6c and 6e). For the second transport path—that is, Path II, these values were −14.9‰ and 13.5‰, respectively (Figs. 6c and 6e). Clearly, the oceanic air mass with low deuterium excess had a relatively more significant impact on the precipitation isotopes in April in the Dongting Lake Basin region" (Line 732-744).

My minor comments are:

Line 21: "Dongting precipitation sourced from ……", should it perhaps read "vapor sourced from ……"?

Response: We followed the comment and have revised "Dongting precipitation sourced from ……" to "vapor sourced from the northern branch of the South Indian Ocean subtropical high" (Line 21).

Line 107: "air dehydration", is this the correct term as used in the manuscript?

Response: We appreciate the reviewer's question and have revisited the description in the reference literature, and indeed it is described as "controlled by the intensity of the air dehydration" in Risi et al. (2010).

Line 115: "are", should this be in the present tense?

Response: We appreciate the reviewer's comments and have revised "are" to "were" (Line 115).

Line 224: "altitude" might be better replaced with "level" for clarity.

Response: We appreciate the reviewer's comments and have revised "altitude" to "level" (Line 225).

Line 226: "data release", is this the appropriate term to use?

Response: We appreciate the reviewer's comments and have revised "data release" to "the release delay days of ERA5" (Line227).

Line 229: It should be noted that the use of "potential height" is preferred, as well as the terms "latitudinal wind" and "meridional wind".

Response: We appreciate the reviewer's comments and have revised the relevant descriptions to "The reanalysis data used in this study include surface pressure ($p_s$,

hPa), potential height of 500 hPa ($H_{500}$, meter), and specific humidity ($q$, kg·kg$^{-1}$), latitudinal wind (m/s), and meridional wind (m/s) at 1000/850/700/600/500/400/300 hPa" (Line 228-231). Moreover, we revised "altitudinal wind" to "latitudinal wind" (Line 230) and have also reviewed the entire manuscript to ensure that the terms "latitudinal wind" and "meridional wind" were used correctly.

Line 259: The order of "$\delta^2H_v$, $\delta^{18}O_v$, $\delta^2H_p$, and $\delta^{18}O_p$" should be adjusted to match the sequence in the text.

Response: We appreciate the reviewer's comments and have revised "$\delta^2H_v$, $\delta^{18}O_v$, $\delta^2H_p$, and $\delta^{18}O_p$" to "$\delta^2H_p$, $\delta^{18}O_p$, $\delta^2H_v$, and $\delta^{18}O_v$" (Line 261) to match the sequence in the text, i.e. "in the precipitation and vertical integral of water vapor ($\delta^2H_p$, $\delta^{18}O_p$, $\delta^2H_v$, and $\delta^{18}O_v$)".

Line 804: Or wherever it may appear, "Dongting precipitation" should be "precipitation in the Dongting Lake Basin".

Response: We appreciate the reviewer's comments and have revised "Dongting precipitation" to "precipitation in the Dongting Lake Basin" (Line 837).

---

## Author Comment (AC5)

*A detailed, point-by-point response to the review comments is given below. Each review comment is repeated followed with* our action to modify the manuscript. *All Page and Line numbers correspond to locations in the revised manuscript.*

Comments from Reviewer #2:

This manuscript presents an analysis of seasonally varying moisture transport paths that influence the water isotopic composition of 1668 days' worth of precipitation samples collected in the Dongting Lake Basin area of China. Transport paths are estimated using a "vector interpolation method" that is applied to vertically integrated moisture fluxes within IsoGSM, a General Circulation Model equipped with water isotopic tracers. The results largely follow expectation. Given that the moisture flux is a function of both wind field and moisture supply, water vapor transport paths do not always align strictly with dominant wind flows. Transport from higher latitudes tends to advect more isotopically depleted water than transport from lower latitudes. And precipitation processes en route to Dongting Lake Basin also influence precipitation isotope ratios. The manuscript is fairly easy to read but lengthy.

Response: We appreciate the comments from the reviewer. We respond to the reviewer's comments as follows:

For the comments "Transport paths are estimated using a "vector interpolation method" that is applied to vertically integrated moisture fluxes within IsoGSM, a General Circulation Model equipped with water isotopic tracers", we utilize the ERA5 reanalysis data to compute the vertical integral of water vapor flux ($Q$), using parameters such as $V$ (vector wind speed), including the latitudinal wind speed ($v$) and

meridional wind ($u$), $q$ (specific humidity), $g$ (acceleration due to gravity), $p_s$ (lower boundary pressure), and $p_t$ (upper boundary pressure), while the ERA5 reanalysis data used in this study include "surface pressure ($p_s$, hPa), the potential height of 500 hPa ($H_{500}$, meter), and specific humidity ($q$, kg·kg$^{-1}$), latitudinal wind (m/s), and meridional wind (m/s) at 1000/850/700/600/500/400/300 hPa" (Lines 234-236). Moreover, we have delineated the pathways of water vapor transport based on the $Q$ field, which represents the water vapor field—that is, as demonstrated in the main text: "In the $Q$ field (Fig. 3a), regarding the Dongting Lake Basin as the endpoint, the vector cluster of the vertical integral of water vapor flux (i.e. the $Q$) directed towards the Dongting Lake Basin delineates the path of water vapor transport in January (black arrow lines in Fig. 3). The water vapor transport path was determined by the rules to find the systematic vapor currents in the $Q$ field, which need to have the same directionality and draw the path along the central axis of the vapor currents. The source regions of water vapor were determined based on the conditions for the formation of air masses, which need to form on a uniform underlying surface and possess stably in terms of isotopic, thermodynamic, and dynamic properties as well as circulation condition, typically located over vast land and ocean regions (Smirnov and Moore, 1999)" (Lines 394-404). Besides, we have analyzed the variations of various factors along the water vapor transport path, drawing data from the isoGSM2 simulations and the ERA5 reanalysis dataset—that is, as demonstrated in the main text "the grid points along the water vapor transport path were identified on the central axis of the path and based on the principle of uniform distribution of the scatter points, and the factors at the grid points were obtained from these scatter points. Besides, the factors at the grid points along the water vapor transport path exhibit, in spatial terms, as average characteristics of conditions over multiple years,

and, in temporal terms, as sequential characteristics of these factors along the water vapor transport path" (Lines 414-420).

Overall, the purpose of our study is to analyze the influences of water vapor sources and transport on the isotopic composition of precipitation in the Dongting Lake Basin. Before our research, no studies had addressed these specific aspects of water vapor origin and isotopic composition, while previous studies' explanations for the variations in precipitation isotopes did not align with the facts of water vapor transport. Although our methodology is fundamental, our interpretation of isotopes is reasonable, and the narrative is clear and coherent. As the comment given by the reviewer "The manuscript is fairly easy to read", primarily because it follows a logical structure throughout.

I have three overarching comments:

First, the study focuses almost entirely on seasonal climatological output from IsoGSM, which feels a bit disappointing given the incredibly large number of daily precipitation samples collected in the Dongting Lake Basin. None of the major findings require the precipitation isotope data, and the paper does not clearly link the IsoGSM interpretation back to the collected samples.

Response: We appreciate the constructive comments from the reviewer. This article primarily focuses on the fundamental seasonal isotopic variations in the Dongting Lake Basin, by comparing actual monitoring data with model simulation data, for instance, precipitation $\delta^{18}O$ ($\delta^{18}O_p$), precipitation deuterium excess (Ex\_$d_p$), and precipitation amount ($P$) measured at the Changsha station and simulated by

isoGSM2 or driven from the ERA5 reanalysis dataset. We found that the isoGSM2 simulated data and ERA5 reanalysis data closely match the actual measurements from the Changsha station, thus confirming the models and reanalysis data to be used in our study. This decision was primarily made because actual monitoring data could not fulfill the spatial and temporal analysis of water vapor transport required for this paper. Additionally, due to space constraints, some research content could not be included in the same article, and it is also impractical for a single paper to encompass too much research material.

Second, the study is largely descriptive in nature. Seasonal maps of moisture, wind, and isotopic output are shown, but the reader is required to eyeball small spatial variations in otherwise large latitudinal gradients. One way to address this might be to plot composite differences (e.g. mapping one season as an anomaly from the annual mean, or plotting the difference between one season and another). Differences between the representative source regions could also be tested statistically, and transport paths could be shown on a single plot—again, to facilitate the reader's ability to compare. The text states that isotopic variations along transport paths are consistent with Rayleigh distillation; however, it does not appear that this was ever tested, which would be quite easy to do in a quantitative manner.

Response: We appreciate the constructive comments from the reviewer. We respond to the reviewer's comments as follows:

Firstly, for the comments "the study is largely descriptive in nature. Seasonal maps of

moisture, wind, and isotopic output are shown, but the reader is required to eyeball small spatial variations in otherwise large latitudinal gradients", we indeed described the results obtained through extensive statistical work, but these data were based on quantitative analysis, as seen in Figs. 4, 6, 8, and 10. For instance, the subtle spatial variations within the otherwise significant latitudinal gradients were captured and displayed in Figs. 3, 5, 7, and 9, while the isotopic variations along the transport paths were included and reflected in Figures 4, 6, 8, and 10.

Secondly, for the comment "One way to address this might be to plot composite differences (e.g. mapping one season as an anomaly from the annual mean, or plotting the difference between one season and another)", our primary focus is on analyzing the factors variations across the four seasons, rather than comparing factors between seasons. However, the comments from the reviewer provided us with an excellent line of thought and represent an innovative aspect for future work. Consequently, we have included this in the main text about the future research "A potential direction for future research could be to investigate the intra-seasonal variations in composite differences across various factors, rather than focusing on inter-seasonal comparisons" (Lines 875-878).

Thirdly, for the comment "and transport paths could be shown on a single plot", given the numerous figures and subplots already presented, we have chosen not to create additional graphs to display this particular element. However, readers can still glean information by comparing the water vapor transport paths depicted in Figs. 3, 5, 7, and 9.

Fourthly, for the comments "The text states that isotopic variations along transport paths are consistent with Rayleigh distillation; however, it does not appear that this was ever tested, which would be quite easy to do in a quantitative manner", we believe that the isotopic variations along the water vapor transport paths encompassed the effects of Rayleigh distillation (i.e. the rainout effect) and water mass balance and isotopic equilibrium. Isolated Rayleigh distillation cannot fully account for the changes in water vapor isotopes and precipitation isotopes along these paths. Therefore, we have added the relevant statement to the main text "In these four months that represent different seasons, variations in the $\delta^{18}O$ and Ex_d of precipitation and water vapor along these water vapor transport paths adhered to principles of Rayleigh fractionation and water balance principles, underscoring the complex transport paths and processes that influence isotopic variations in precipitation in the Dongting Lake Basin" (Lines 860-864). Furthermore, the factors along the water vapor transport paths include both static spatial averages (i.e. the point averages) and the temporal sequence along the paths, and we have also made relevant statements in the main text regarding this aspect "the factors at the grid points along the water vapor transport path exhibit, in spatial terms, as average characteristics of conditions over multiple years, and, in temporal terms, as sequential characteristics of these factors along the water vapor transport path" (Lines 417-420).

Third, it is not clear to me what new information about water cycle processes (including transport) this study provides, other than to provide a thorough description

of seasonal mean wind flows, moisture transport paths, and vapor and precipitation isotope ratios around a single location. It motivates me to ask: was the Dongting Lake Basin chosen for a particular reason as a scientifically important location? Or was this a study of opportunity based on the large number of event-based precipitation samples?

Response: We appreciate the constructive comments from the reviewer. We respond to the reviewer's comments as follows:

For the comment "it is not clear to me what new information about water cycle processes (including transport) this study provides, other than to provide a thorough description of seasonal mean wind flows, moisture transport paths, and vapor and precipitation isotope ratios around a single location", our study deliberately emphasized not the water cycle itself, but rather the transport of water vapor and its influence on the isotopic composition of precipitation.

For the comment "It motivates me to ask: was the Dongting Lake Basin chosen for a particular reason as a scientifically important location?", we believe that the Dongting Lake Basin, situated in the East Asian monsoon climate zone, offers a rather typical representation for research purposes. A significant amount of work has already been conducted within this basin, including sampling and observation of various water bodies, and analysis of the stable isotopes in stalagmites, lake sediments, and peat. This study is part of a series of research efforts in the Dongting Lake Basin.

For the comment "Or was this a study of opportunity based on the large number of event-based precipitation samples?", However, from a philosophical perspective, we

believe that universality is embodied within particularity. By examining the seasonal variations in water vapor transport and their influences on precipitation isotopes in the Dongting Lake Basin, we can gain a broader spatial understanding of the scientific issues surrounding water vapor transport. Therefore, we added the relevant statement in the main text "Focusing on the Dongting Lake Basin within the East Asian monsoon area, and drawing upon fundamental theories of meteorology, water vapor diagnostics, and water vapor calculations, a broader spatial understanding of the scientific issues surrounding water vapor transport can be achieved" (Lines 153-156)

I feel that this study would benefit from some additional context for why this work was conducted, how the IsoGSM output guides us in understanding the precipitation data, and the inclusion of quantitative analyses.

Response: We appreciate the constructive comments from the reviewer. As demonstrated in our previous responses, the data we employed comprises the isoGSM2 simulated data and ERA5 reanalysis data. The variations in $Q$, $P$, and isotopes in water vapor and precipitation along the transport paths can be regarded as quantitative changes and subject to quantitative analysis.

I also have two additional comments specific to methodology:

First, the study needs to provide more information about the "vector interpolation method." The introduction criticizes other moisture transport evaluation approaches, such as those based on Langrangian back trajectory techniques, because these cannot

definitively determine what moisture becomes rain. However, it is not clear to me how interpreting vertically integrated moisture flux fields in IsoGSM would allow one to do this either. In fact, there is a possibility that the vertically integrated transport could be quite distinct from the transport vectors of moist layers that generate precipitation (e.g. if free tropospheric moisture convergence is more critical than low-level moisture convergence for producing rain).

Response: We appreciate the constructive comments from the reviewer. For the comments "the study needs to provide more information about the "vector interpolation method", the method we employ is a fundamental approach in meteorological plotting: using the center of the Dongting Lake Basin as the endpoint and aligning parallel to the systematic transport of the $Q$ vector cluster, which can be described as following the direction of the water vapor streamline. Therefore, in the main text, we introduce this concept as follows: "In the Q field (Fig. 3a), regarding the Dongting Lake Basin as the endpoint, the vector cluster of the vertical integral of water vapor flux (i.e., the Q) directed towards the Dongting Lake Basin delineates the path of water vapor transport in January (black arrow lines in Fig. 3)" (Lines 394-397).

For the comment "The introduction criticizes other moisture transport evaluation approaches…….convergence for producing rain", HYSPLIT is capable of depicting water vapor transport paths at various levels, yet this does not necessarily correspond to the water vapor responsible for precipitation. This finding has been explicitly stated in the referenced literature. In our results and analysis, the variations of $Q$ and $P$ along the transport pathways, as well as isotopic changes, can be considered quantitative

alterations. Moreover, the ratio of *P* (precipitation amount) to *PW* (precipitable water) can represent both the rate of precipitation generation and the level of rainout, which has been reflected in the paper previously published by our research group. In this study, we focused on specific precipitation days and analyzed the impact of different precipitation forms, such as convective and advective precipitation, on the isotopic composition of precipitation in the study area. However, due to space limitations, not all content could be presented in this submitted manuscript. The reference is as follows:

Xiao, Z., Zhang, X., Xiao, X., Chang, X., & He, X. (2024). The Effect of Convective/Advective Precipitation Partitions on the Precipitation Isotopes in the Monsoon Regions of China: A Case Study of Changsha. Journal of Hydrometeorology, 25(4), 581-590.

Second, it appears that the paper weights all days equally in its analysis, regardless of whether rain occurred or not. Thus it is hard to draw conclusions about whether the descriptive analysis accurately describes the conditions in which precipitation samples were collected.

Response: We appreciate the constructive comments from the reviewer. In fact, our analysis represents the average results over multiple years from January 1979 to December 2017, totaling 468 months, including monthly precipitation amount ($P$, mm), stable isotopes ($\delta^2H$ and $\delta^{18}O$) in the precipitation and vertical integral of water vapor ($\delta^2H_p$, $\delta^{18}O_p$, $\delta^2H_v$, and $\delta^{18}O_v$), and the calculated deuterium excess in water

vapor and precipitation (Ex_d$_v$ and Ex_d$_p$). Therefore, our analysis is based on the multi-year averages of these factors to characterize the long-term conditions for four representative months, and we examine the impact of water vapor transport paths on the isotopic composition of precipitation in the Dongting Lake Basin. Furthermore, addressing the author's concerns, the relevant content has been demonstrated in previously published papers by our research group. In that study, we focused on specific precipitation days, analyzing the effects of different precipitation forms—convective and advective precipitation—on the isotopic composition of rainfall. The reference is as follows:

Xiao, Z., Zhang, X., Xiao, X., Chang, X., & He, X. (2024). The Effect of Convective/Advective Precipitation Partitions on the Precipitation Isotopes in the Monsoon Regions of China: A Case Study of Changsha. Journal of Hydrometeorology, 25(4), 581-590.

Other line-by-line comments:

Line 53 - This mention of isotope paleoclimate applications seems out of context. Could more information be provided or this sentence eliminated?

Response: We followed comments from the reviewer and eliminated this sentence.

Line 89 - Here and in all other instances, I believe "southwesterly" is meant, rather than southwestward.

Response: We appreciate the constructive comments from the reviewer. However, our

use of "southwesterly" is intended to indicate that the wind originates from the southwest, not that it is blowing towards the southwest. In other words, "wind from the northwest direction" and "northwesterly winds" convey the same meaning. To revise this would contradict our original intent.

Line 94 - I do not understand what this concluding sentence is trying to say.

Response: We apologize for the unclear statement and revised this sentence to "Despite the presence of water vapor input from the northwest direction, we observe no correlation between the transport of water vapor from this direction and the regional water vapor budget or precipitation amount. In some instances, an inverse relationship is even apparent. These relationships imply that northwesterly winds do not exert a direct influence on the precipitation generation in the region" (Lines 90-94).

Line 116 - Rain forms from water vapor that condenses, so I'm not sure how water vapor source regions are not relevant for or the same as precipitation source regions, unless the source regions are being defined climatologically (irrespective of whether there is rain). Clarification is needed.

Response: We appreciate the constructive comments from the reviewer. For the determination of the source regions, we demonstrated that "The source regions of water vapor were determined based on the conditions for the formation of air masses, which need to form on a uniform underlying surface and possess stably in terms of

isotopic, thermodynamic, and dynamic properties as well as circulation condition, typically located over vast land and ocean regions (Smirnov and Moore, 1999)" (Lines 401-403). What we intend to convey is that the source regions of water vapor may not be the primary factor directly affecting the precipitation in the Dongting Lake Basin, rather than suggesting that "water vapor source regions are not relevant for or the same as precipitation source regions". Therefore, we have demonstrated that "With emphasis, for the water vapor source of precipitation in the Dongting Lake Basin, the oceanic representative regions located at low latitudes may not necessarily be the initial water vapor source regions, and the relationship between upstream and downstream regions may not entirely be point-to-point, as there were continuous water recycling and rainout processes along the water vapor transport path (Pokam et al., 2012; Risi et al., 2013; Christner et al., 2018)" (Lines 726-731) and "In the process of seasonal changes, as air masses move out of their source regions, their physical and weather characteristics also change with the variations in underlying surface properties and large-scale vertical motion conditions" (Lines 773-776).

Line 129 - I'm not sure what is meant by "by simple deduction." What follows is not obvious to me.

Response: We apologize for the unclear statement. What we intend to convey is that the water vapor transport during the summer monsoon aligns with the spatial distribution of precipitation isotopes under the influence of latitudinal and continental effects and is consistent with the Rayleigh distillation principle for water stable

isotopes, however, water vapor transport during the winter monsoon does not follow the above spatial distribution and Rayleigh distillation principle. Therefore, we have demonstrated that "Typically, during the summer monsoon, prevailing southeast or southwest winds dominate the East Asian monsoon region, with water vapor for precipitation originating from low-latitude oceans (Barker, et al., 2015; Wu et al., 2015; Tang et al., 2015), while precipitation isotopes are significantly depleted influenced by intense rainout effects along the water vapor transport paths during this period (Zhou et al., 2019; Wu et al., 2022). Conversely, during the winter monsoon, northwest or northeast winds prevail in the East Asian monsoon region, the precipitation isotopes should be more enriched if water vapor for precipitation is carried by westerlies or originates from the evaporation of inland regions (e.g., Liu et al., 2011; Wu et al., 2015; Shi et al., 2021). However, both actual observations from the Global Network of Isotopes in Precipitation (GNIP) and simulations from isotope-enabled General Circulation Models (isoGCMs) consistently demonstrate that, whether during the summer or winter monsoon, the spatial distribution of precipitation isotopes in the East Asian monsoon region exhibits significant latitudinal and continental effects—that is, the precipitation isotopes become more depleted with the increases of latitude or distance from the ocean (Feng et al., 2009; Zhang et al., 2012; Zhang et al., 2016)." (Lines 127-142).

Line 139 - This seems to be the problem statement for the paper: what the analysis will address. This should appear sooner in the introduction, and the analyses should

test these ideas (e.g. test consistency with Rayleigh distillation).

Response: We appreciate the constructive comments from the reviewer and added the relevant introduces in the head of this paragraph "Verifying whether the stable isotopic composition of water vapor undergoes changes consistent with Rayleigh distillation during transport, and assessing the impact of this transport on the isotopic composition of regional precipitation, constitute significant research objectives" (Lines 120-123).

Line 155 - Sub "influence" for "influencing."

Response: We followed the comment from the reviewer and revised this sentence to "reveal the mechanisms by which water vapor sources and transport paths in the monsoon region influence precipitation amounts and isotopes" (Lines 159-161).

Line 223 - I'm not sure what is meant by the "fractionation process…cannot be directly observed." We cannot observe individual molecules evaporating and condensing, but we do observe the partitioning of isotopes between distinct water phases.

Response: We apologize for the unclear statement and revised this sentence to "Since the direct observation of the isotopic fractionation process in the atmosphere is extremely challenging, analyzing the variations of atmospheric stable isotopes requires the application of stable isotopes fractionation theory along with the fundamental principles and methods of meteorology" (Lines 240-243).

Line 243 - What are "the driving factors" referring to?

Response: We apologize for the unclear statement and revised this sentence to "The driving data include sea surface temperature, sea ice, and temperature and horizontal wind fields in 28 vertical layers" (Lines 250-252).

Line 285 - Here and elsewhere, variables should be defined. Deuterium excess is never explicitly defined in the paper, and the notation is irregular. Typically, we would write d for deuterium excess.

Response: We apologize for the mistake and explicitly defined Deuterium excess in the paper when it first occurred—that is, "The water stable isotope simulation data used in this study are from isoGSM2 (January 1979 to December 2017, totaling 468 months), including monthly precipitation amount ($P$, mm), stable isotopes ($\delta^2H$ and $\delta^{18}O$) in the precipitation and vertical integral of water vapor ($\delta^2H_p$, $\delta^{18}O_p$, $\delta^2H_v$, and $\delta^{18}O_v$), and the calculated deuterium excess in water vapor and precipitation (Ex_d$_v$ and Ex_d$_p$)" (Lines 263-267).

Figure 2 Caption is missing an "E" before "RA5."

Response: We apologize for the mistake and replace "RA4" with "ERA5" (Line 310).

Line 359 - "Lied" is not the correct word.

Response: We apologize for the mistake and revised this sentence to "Unlike most

regions of the East Asian continent, the Dongting Lake Basin was situated on a wet tongue, benefiting from the Southwest Vortex in the eastern Tibetan Plateau" (Lines 364-366).

Line 360 - it would help to point the vortex out if it is significant for the interpretation.

Response: We followed the comment from the reviewer and added more relevant descriptions about the Southwest Vortex "This Southwest Vortex is a cyclonic bypass flow of westerlies from the southern branch of the Tibetan Plateau, as this vortex moves eastward with the westerly belt, it brings precipitation to the downstream areas" (Lines 366-369).

Line 381 - It is a bit misleading to state that deuterium excess is generally affected by condensation. It is often conserved when condensation occurs under thermodynamic equilibrium (at saturation). It is not when condensation occurs under supersaturation. There is also a dependence on temperature conditions, but this is more detail than required for this sentence.

Response: We appreciate the constructive comments from the reviewer and revised this sentence to "Typically, the $Ex\_d_p$ largely depended on the $Ex\_d_v$, but processes such as condensation and super-saturation in ice-water mixed clouds, secondary evaporation below clouds, evaporation from underlying surfaces, and the exchange and diffusion of water vapor isotopes could cause precipitation isotopes to deviate to

varying degrees from atmospheric water vapor isotopes (Zhang et al., 2016)" (Lines 388-393).

Line 393 - It is very hard to see the January wind vectors to verify this statement.

Response: We apologize for the unclear statement and added the relevant descriptions as "It can be seen that this water vapor transport path was not consistent with the prevailing wind direction in January as shown in Fig. 1a, which was represented by the black arrows with northwesterly winds prevailing in the Dongting Lake Basin as shown by the average wind field at the 850 hPa" (Lines 407-411).

Line 401 - here and elsewhere, "relatively positive" is misleading, since these values are always negative. "Less negative" or "higher" could work instead.

Response: We appreciate the comments from the reviewer and revised "relatively positive" to "higher" (Lines 425).

Figure 5 - It should be made more clear here and in the text that there are two dominant transport paths for this particular month. At first, I thought the January path was being copied over from the preceding plots. This leads me also to ask: how are two dominant paths selected? How does the method permit the identification of more than one average path?

Response: We appreciate the comments from the reviewer. We determined the dominant transport paths based on the flow of water vapor in the water vapor field (i.e.

the $Q$ field), which can be described as the "water vapor rivers" indicated by the vector lengths and directions of $Q$ in Fig. 5a. Based on this analysis, we identified two distinct dominant transport paths in April.

Line 440 - I'm not sure one can make such a broad conclusion based solely on the fact that there are similar deuterium excess values between the lake basin and the ocean. Also, does the large bin size on the color scale of the maps hide small-scale spatial variability?

Response: We appreciate the comments from the reviewer. However, we have roughly deduced the relationship between the water vapor sources and the Dongting Lake Basin by comparing the deuterium excess and stable isotopic values between the basin and the ocean. This is a relatively coarse and broad estimation method, yet it is widely utilized in related isotopic hydrometeorological studies.

Line 487 - What is the "variation rule?" I am unfamiliar with this concept. A brief description would help.

Response: We apologize for the unclear statement and revised this sentence to "following the variation rule of deuterium excess during water vapor transport—that is, as the rainout effect progressed, the heavier isotopes preferentially left the air parcel or cloud during the water vapor transport processes and generated precipitation, thus resulted in subsequent precipitation having increasingly higher deuterium excess values (Vasil'chuk, 2014)" (Lines 510-514).

Line 557 - I'm not sure I agree without knowing more. Similar advective paths is one reason deuterium excess may be similar between sites. They could also be influenced by similar degrees of sub-cloud evaporation or other processes that produce similar signals.

Response: We appreciate the constructive comments from the reviewer and added the relevant statement in the manuscript: "However, there were no significant changes in both the Ex_$d_v$ and Ex_$d_p$ (Figs. 8e and 8f), the reasons may be due to the continuous water vapor supply from low-latitude oceans, similar advective paths between the grid points, and the similar degrees of sub-cloud evaporation" (Lines 582-586).

Line 638 - Should the representative regions be interpreted as the vapor source regions? This is not clear to me.

Response: These regions are located over the ocean, essentially forming the water vapor source areas. We have revised this statement following the reviewer's comments: "the water vapor regions corresponding to the Arabian Peninsula (40°E~56°E, 16°N~28°N), the Arabian Sea (56°E~74°E, 10°N~20°N), the Bay of Bengal (80°E~98°E, 8°N~18°N), and the western Pacific Ocean (120°E~160°E, 6°N~20°N), along with the inland regions of Dongting Lake Basin (110°E~114°E, 25°N~30°N) and East Asia monsoon region (110°E~135°E, 42°N~55°N) were labeled as Regions I, II, III, IV, V, and VI, respectively (Fig. 11)" (Lines 665-671).

Line 654 - I don't think "seasonal differences" is the right term. I believe the text is describing the difference between the max and min values for each season, not differences between seasons, which is what the former implies. Perhaps "seasonal range?" It's also not clear to me why this particular metric is chosen.

Response: We appreciate the constructive comments from the reviewer and revised "seasonal differences" to "seasonal range" (Lines 683, 686,689, and 690).

Table 1 - I think some of the Discussion length could be cut down by removing parts of the text that simply repeat what is already in Table 1.

Response: We followed the comment and removed these sentences from the text: "According to the statistics in Table 1, in January, the average $\delta^{18}O_v$ and Ex_$d_v$ were −19.04‰ and 18.45‰, respectively, in Region I under the latitudinal water vapor transport, while −18.82‰ and 15.21‰, respectively, in Region V with their differences only 0.22‰ and 3.24‰, respectively; In April, the average $\delta^{18}O_v$ and Ex_$d_v$ were −16.22‰ and 16.93‰, respectively, in Region I also under the latitudinal water vapor transport, while −17.94‰ and 13.91‰, respectively, in Region IV under the meridional water vapor transport, and −14.91‰ and 13.55‰, respectively, in Region V; In June, the average $\delta^{18}O_v$ were −14.45‰ and −18.10‰, respectively, the average Ex_$d_v$ values were 13.09‰ and 13.98‰, respectively, in Regions II and III, all under the meridional water vapor transport, while the average $\delta^{18}O_v$ and average Ex_$d_v$ were −20.77‰ and 15.03‰, respectively, in Region V after experiencing intense rainout processes; In October, the average $\delta^{18}O_v$ and Ex_$d_v$ were −22.30‰

and 17.72‰, respectively, in Region IV under the weakened meridional water vapor transport, showing non-significantly differences from the values of −22.09‰ and 17.14‰, respectively, in Region V (Table 1)".

Line 683 - Sub "Furthermore" for "Furtherly"

Response: We followed the comment and replaced "Furtherly" with "Furthermore" (Line 707).

Line 744 - I'm not sure what a "modificatory" air mass is or why it should become more negative. Condensation and precipitation cause air masses to lose heavy isotopes, as does mixing.

Response: We apologize for the unclear statement and added the relevant descriptions about the modificatory air mass "In the process of seasonal changes, as air masses move out of their source regions, their physical and weather characteristics also change with the variations in underlying surface properties and large-scale vertical motion conditions. East Asia is primarily controlled by modificatory air masses, which were commonly used to describe air masses that have changed as they move through different regions (Ding, 1990; Chang et al., 2012)" (Lines 773-778). The modificatory air mass becomes more depleted in heavier isotopes than the original air mass, a phenomenon that can be explained as "following the variation rule of deuterium excess during water vapor transport—that is, as the rainout effect progressed, the heavier isotopes preferentially left the air parcel or cloud during the

water vapor transport processes and generated precipitation, thus resulted in subsequent precipitation having increasingly higher deuterium excess values (Vasil'chuk, 2014)" (Lines 510-514). Therefore, we revised the relevant statements to "Whether cold and dry air masses moving southward or warm and moist air masses moving northward, the isotopic composition of water vapor in the modificatory air mass continues to become more negative, while the deuterium excess of water vapor continues to become more positive than the original air mass, following the variation rule of stable isotope and deuterium excess during water vapor transport (Vasil'chuk, 2014; Zhou et al., 2019; Xu et al., 2019; Jackisch et al., 2022)" (Lines 778-784).

Line 749 - That air masses are distinct in various ways can be taken for granted. Hopefully we can conclude more than this from this original work? A more specific conclusion sentence would be welcome.

Response: We followed the comment and made more specific conclusion sentences "In summary, based on the comparison of stable isotopes of water vapor and precipitation in different seasons and representative regions presented in this study, it can be observed that, as an integral component of the climate system, air masses in various regions not only exhibit differences in thermodynamic, dynamic, hydrous, and static properties, but are also influenced by interactions between air masses, the underlying surface, and the intensity of convection, among others" (Lines 786-792).

Line 783 - Point B in the schematic is where mixing between cold and warm advected

air masses should occur, and yet mixing is not discussed in the paper as a possible process shaping the water vapor and precipitation isotope ratios. Some discussion of the contribution of mixing might be a worthwhile addition depending on what a revised, more quantitative analysis yields.

Response: We appreciate the constructive comments from the reviewer and added more detailed descriptions about Point B: "However, at point B located within the front zone, the wind direction and speed are uncertain (Fig. 12b). Specifically, this front zone marked the transition from a warm air mass to a cold one, or vice versa, where meteorological factors have undergone rapid changes. Mixing between cold and warm advected air could occur within this zone, manifesting as a shear zone in wind fields, or as alternating southerly and northerly winds" (Lines 824-829).

---

## Author Response (AR2)

*A detailed, point-by-point response to the review comments is given below. Each review comment is repeated followed with our action to modify the manuscript. All Line numbers correspond to locations in the revised manuscript.*

**Editor decision from Peter Haynes:**

There were three reviewers for this paper, two anonymous and one non-anonymous. One of the anonymous reviewers (1)-- who provided a second report -- and the non-anonymous reviewer -- whom I consulted off-line to save time -- were satisfied with your revisions and recommend publication at this stage. The other anonymous reviewer (2) was more critical and considered that the paper still has significant weaknesses. Having looked carefully at reviewer 2's report and your revised paper, I feel that they make some good points. One concern is clarity of the methodology -- for example the method you use to identify the 'mean vapor transport path' -- you refer to 'rules' but what where these rules? If there is an algorithm here then it should be given explicitly. If in fact there is no algorithm then that should be stated, some kind of statement should be given about how the paths were drawn and it should be admitted that there is some arbitrariness in this choice. The reviewer makes a similar point about source regions.

The general point here, as emphasised by the reviewer, is that the work should reproducible by others. If a non-algorithmic choice has been made then the basis for that choice should be clearly stated.

Response: Thank you very much for your helpful comments.

In our study, the water vapor transport paths were determined by identifying systematic vapor currents in the $Q$ field (the vertical integral of water vapor flux). These paths were delineated along the central axis of the vapor currents, ensuring consistent directionality. While this approach involves some subjectivity, it is guided by specific criteria. Moreover, the source regions of water vapor were determined based on the conditions for the formation of air masses, which require a uniform underlying surface and stable isotopic, thermodynamic, and dynamic properties, as well as favorable circulation conditions. These regions are typically located over vast

land and ocean areas and serve as the starting points of the water vapor transport paths. Our identified water vapor transport paths are conceptually similar to atmospheric rivers, which are long, narrow corridors of strong horizontal water vapor transport typically associated with low-level jets (Ralph et al., 2017; Payne et al., 2020). However, our paths are derived from the climatological mean state (multi-year monthly averages) rather than short-term events, which is a key distinction from atmospheric rivers that generally last for a few days to a week (Dettinger et al., 2013). We acknowledge that while our method is based on scientific criteria, there is some degree of empiricism and subjectivity in defining these paths and source regions. However, this approach ensures that we capture the systematic vapor currents that have the most significant influence on local precipitation and its isotopic composition. We believe this provides a reasonable basis for identifying the dominant vapor transport paths and the primary sources of water vapor contributing to precipitation in the Dongting Lake Basin.

We revise the manuscript to clarify these points and provide a more transparent explanation of our methodology: "The water vapor transport paths were determined by identifying systematic vapor currents in the $Q$ field, which need to have the same directionality and draw the path along the central axis of the vapor currents. Our identified water vapor transport paths are conceptually similar to atmospheric rivers, which are long, narrow corridors of strong horizontal water vapor transport typically associated with low-level jets (Ralph et al., 2017; Payne et al., 2020). However, our paths are derived from the climatological mean state (multi-year monthly averages) rather than short-term events, which is a key distinction from atmospheric rivers that generally last for a few days to a week (Dettinger et al., 2013). The source regions of water vapor were determined based on the conditions for the formation of air masses, which need to form on a uniform underlying surface and possess stability and similarity in terms of isotopic, thermodynamic, and dynamic properties as well as circulation conditions (Smirnov and Moore, 1999). These regions are typically located over vast land and ocean areas and serve as the starting points of the water vapor transport paths. Although there is some empiricism and subjectivity, there is no

explicit algorithm to determine the exact water vapor transport path and source regions of water vapor. However, this approach is based on certain criteria and ensures that we capture the systematic vapor currents that have the most significant influence on the local precipitation and its isotopic composition, and it provides a reasonable basis for identifying the dominant vapor transport directions and the primary sources of water vapor contributing to precipitation in the Dongting Lake Basin" (Line 395-415).

For a detailed exploration of these aspects, please refer to the previous publications:

Dettinger, M. D. (2013). Atmospheric rivers as drought busters on the US West Coast. Journal of Hydrometeorology, 14(6), 1721-1732.

Payne, A. E., Demory, M. E., Leung, L. R., Ramos, A. M., Shields, C. A., Rutz, J. J., ... & Ralph, F. M. (2020). Responses and impacts of atmospheric rivers to climate change. Nature Reviews Earth & Environment, 1(3), 143-157. https://doi.org/10.1038/s43017-020-0030-5

Ralph, F. M., Dettinger, M. D., Cairns, M. M., Galarneau, T. J., & Eylander, J. (2018). Defining "atmospheric river": How the Glossary of Meteorology helped resolve a debate. Bulletin of the American Meteorological Society, 99(4), 837-839. https://doi.org/10.1175/BAMS-D-17-0157.1

The other two major points made by the reviewer seem valid to me. (The first is asking what has been determined by related work in a different publication and what has not. The second is simply asking for clarification.)

Response: We appreciated the helpful comments from the editor and reviewer.

**For the second major issue made by the reviewer, it is mainly related to the time scale of the different studies**. In our manuscript, we focus on the climatological mean state of atmospheric processes over four representative months (January, April, June, and October, which representing winter, spring, summer, and autumn, respectively) from 1979 to 2017. Our analysis is based on the monthly average values of vertical integral water vapor flux $Q$, precipitation amount $P$, and isotopic compositions of water vapor and precipitation. The goal is to reflect the general

characteristics of water vapor transport paths and their influence on precipitation in the Dongting Lake Basin for each month, rather than focusing on individual precipitation events. This approach is valid because the magnitude of water vapor flux $Q$ is directly related to precipitation amounts $P$. For example, in January and October, the low values of $Q$ correspond to low precipitation amounts $P$ (Figs. 4 and 10), indicating that the overall transport conditions are less conducive to precipitation formation. Therefore, our analysis does not require differentiation between precipitating and non-precipitating days, as we are examining the climatological mean state for each month. In contrast, our previous study (Xiao et al. 2024, DOI: 10.1175/JHM-D-23-0084.1) focused on the daily scale and specifically examined the relationship between precipitation isotopes and precipitation types during precipitation events. This study required differentiation between precipitating and non-precipitating events to understand the influence of convective and advective processes on isotopic compositions. However, this previous work does not conflict with the current manuscript. Instead, it complements our analysis by providing insights into the processes occurring during individual precipitation events, while our current study examines the broader climatological context.

Therefore, we followed the comments and added the relevant statements in the manuscript: "In this study, we focus on the climatological mean state of water vapor transport and its influence on precipitation in the Dongting Lake Basin over four representative months (January, April, June, and October, which representing winter, spring, summer, and autumn, respectively) from 1979 to 2017. Our analysis is based on the monthly average values of vertical integral water vapor flux $Q$, precipitation amount $P$, and isotopic compositions of water vapor and precipitation. This approach reflects the general characteristics of water vapor transport paths and their influence on precipitation for each month, rather than focusing on individual precipitation events. The magnitude of water vapor flux $Q$ is directly related to precipitation amounts $P$, and thus, our analysis does not require differentiation between precipitating and non-precipitating days. For example, low values of water vapor flux $Q$ in January and October correspond to low precipitation amounts, indicating that the

overall transport conditions are less conducive to precipitation" (Line 787-799).

**For the third major issue made by the reviewer**, we apologize for the unclear statement and have clarified the water vapor isotope ratios: "The water stable isotope simulation data used in this study are from isoGSM2 (January 1979 to December 2017, totaling 468 months), including monthly precipitation amount ($P$, mm), stable isotopes ($\delta^2H$ and $\delta^{18}O$) in the precipitation ($\delta^2H_p$, $\delta^{18}O_p$), the vertical integral of water vapor isotopes ($\delta^2H_v$, and $\delta^{18}O_v$) of 17 pressure levels from 1000 hPa to 10 hPa, and the calculated deuterium excess in water vapor and precipitation ($d_v$ and $d_p$)" (Line 263-268).

The referee makes other good points of detail. In particular I have some sympathy with their view that whilst the abstract states 'In different seasons, the variations in water stable isotopes along water vapor transport paths adhered to Rayleigh fractionation and water balance principles.' -- it is difficult how exactly this paper verifies that. Perhaps you simply mean that there some agreement between the model (which incorporates Rayleigh fractionation and water balance principles) and the observations e.g. as shown in your Figure -- but has your paper provided any new evidence on this point. I can't see any detailed analysis of the relative importance of these effects? The seasonal variation could result from many different effects.

Response: We appreciate the helpful comments from the reviewer and editor. We apologize for any confusion caused by our wording and appreciate your feedback. You are correct that our study does not provide direct evidence to verify that the variations in water stable isotopes along water vapor transport paths strictly adhere to Rayleigh fractionation and water balance principles. **Instead, our findings show that there is some agreement between the model (which incorporates Rayleigh fractionation and water balance principles) and the observations. This agreement suggests that the model is capable of capturing the general trends in isotopic variations, but we acknowledge that this does not constitute direct proof of the underlying mechanisms.**

**We have revised the abstract and the relevant sections of the manuscript to**

**clarify this point and avoid any misleading implications.** Following the comments and suggestions, we have revised the related sentences "In different seasons, the variations in water stable isotopes along water vapor transport paths adhered to Rayleigh fractionation and water balance principles", "Verifying whether the stable isotopic composition of water vapor undergoes changes consistent with Rayleigh distillation during transport, and assessing the impact of this transport on the isotopic composition of regional precipitation, constitute significant research objectives", and "In these four months that representing different seasons, variations in the $\delta^{18}O$ and deuterium excess of precipitation and water vapor along these water vapor transport paths adhered to principles of Rayleigh fractionation and water balance principles, underscoring the complex transport paths and processes that influence isotopic variations in precipitation in the Dongting Lake Basin" to "In different seasons, the variations in water stable isotopes along water vapor transport paths show some agreement with Rayleigh fractionation and water balance principles, as reflected in the model simulations and observations" (Line 25-28), "Assessing whether the stable isotopic composition of water vapor shows changes consistent with Rayleigh distillation during transport, and evaluating the impact of this transport on the isotopic composition of regional precipitation, are important research objectives" (Line119-122), and "In these four months representing different seasons, variations in the $\delta^{18}O$ and deuterium excess of precipitation and water vapor along these water vapor transport paths show some agreement with Rayleigh fractionation and water balance principles. This highlights the complex transport paths and processes that influence isotopic variations in precipitation in the Dongting Lake Basin" (Line 923-927).

Having looked more carefully at the paper myself, a specific question is what is the lower axis in Figure 4 and similar Figures supposed to represent? What do the numbers mean? I can't find any explanation of that. (In general I felt that Figure captions should be clearer -- the reader should be told in the caption what they are seeing -- they should not have to guess.

Response: We appreciate the helpful comments from the reviewer and editor and apologize for any confusion caused by the lack of explanation in the captions. **The lower axis in Fig. 4 and similar figures represent the grid points along the water vapor transport path from the source region to the endpoint (i.e., the Dongting Lake Basin). The numbers on this axis indicate the serial numbers corresponding to the grid points from the source region to the Dongting Lake Basin along the water vapor transport path, which were selected at almost equal intervals**. At each of these grid points, we derived the variations of six factors ($Q$, $P$, $\delta^{18}O_v$, $\delta^{18}O_p$, $d_v$, and $d_p$) from the corresponding factor fields. These points help us analyze the changes in these factors along the transport path.

For the aims to ensure that readers can easily understand the information presented, we have revised the figure caption and main text to provide clearer explanations as "Fig. 4 Mean variations of $Q$ (a), $P$ (b), $\delta^{18}O_v$ (c), $\delta^{18}O_p$ (d), $d_v$ (e), and $d_p$ (f) along the vapor transport path in January. The numbers at the lower axis represent the serial numbers corresponding to the grid points along the water vapor transport path from the source region to the Dongting Lake Basin, and the points along the path were selected at almost equal intervals to capture the variations of each factor, hereinafter the same" (Line 450-454) and "In analyzing the variations along the water vapor transport paths, we selected several points along the path from the source region to the Dongting Lake Basin, spaced at nearly equal intervals. Six series of factors at the grid points along the water vapor transport path were derived from each factor field in January, including the variations of $Q$, $P$, $\delta^{18}O_v$, $\delta^{18}O_p$, $d_v$, and $d_p$, these points allow us to examine the changes in these factors along the transport path. Moreover, the grid points along the water vapor transport path were identified on the central axis of the path and based on the principle of uniform distribution of the scatter points, and the factors at the grid points were obtained from these scatter points. Besides, the factors at the grid points along the water vapor transport path exhibit, in spatial terms, average characteristics of conditions over multiple years, and, in temporal terms, sequential characteristics of these factors along the water vapor transport path" (Line 430-441).

In summary, my view is that, whilst reviewer 2's views and comments are different from the other reviewers, they are valid and you should consider them further. Please provide clear responses to the reviewer comments and an appopriately revised version of the paper and I will then consider the paper further.

Response: We appreciate the helpful comments from the reviewer and have revised the manuscript accordingly, the details can be found in the responses to the specific concerns.

**Comments from Anonymous Referee #1:**

Upon my thorough review of the revised manuscript submitted by the authors, I am pleased to report that the paper has undergone significant and commendable improvements. The manuscript primarily investigates the water vapor sources from the perspectives of meteorology and isotope hydrology, a field of study that is both routine and critical, particularly within meteorological departments. The seasonal differences in water vapor sources are a major cause of the seasonal variations in the stable isotopes of precipitation. Therefore, a correct understanding and recognition of water vapor sources and transport are of paramount importance for accurately assessing the water cycle and implementing effective precipitation forecasting. As the authors have mentioned, and as my understanding aligns, previous studies have often favored the use of HYSPLIT or wind field diagrams for water vapor tracing. However, HYSPLIT and wind fields typically indicate the movement of air particles under the influence of wind forces or else forcing factors, which may not be effective sources of precipitation, especially when the air masses are dry. For instance, in the East Asian monsoon region, the prevailing northerly winds during winter bring continental air masses that are cold and dry, thus not conducive to effective precipitation. Many geographical researchers conducting atmospheric precipitation isotope source studies, often lack a meteorological background and may overlook this pattern. The current paper addresses this gap by analyzing the water vapor sources and their impact on regional precipitation isotopes based on the water vapor flux field. It reveals that the

isotopic abundance of precipitation is influenced by the rainout effect along the water vapor transport path, en route water vapor replenishment, and the isotopic compositions of the source region. This analysis adheres to both the water mass balance and the stable isotope balance. The paper offers valuable insights and references in both methodology and conclusions. Overall, the manuscript employs a rich dataset, employs appropriate computational methods, and demonstrates a certain level of innovation and broad inspiration. I recommend the publication of this paper.

Response: We appreciate the positive comments from the reviewer and the reviewer's recommendations for this manuscript. Additionally, we have revised the format of the references in the manuscript to comply with the reference format requirements of ACP, as requested by another editor.

**Comments from Anonymous Referee #2:**

I remain concerned about two central aspects of this work, one related to the methodology and the other related to the study design.

1. The methodology remains unclear, and consequently the science is not reproducible.

The revision states: "The water vapor transport path was determined by the rules to find the systematic vapor currents in the $Q$ field, which need to have the same directionality and draw the path along the central axis of the vapor currents." This is not clear. What rules do these refer to? My suspicion is that the paper simply identifies dominant patterns of moisture fluxes by eyeballing the moisture flux fields (e.g. looking at the qualitatively coherent direction of the vectors). This may be sufficient, but it is worth clarifying if this is what is actually done, especially now that we can explicitly tag moisture sources in simulations. Would we expect the transport paths identified in this work to agree with sources identified by tags? Would they agree with particle dispersion models or back trajectory models? Without clarification regarding what the "systematic vapor currents" are or what "rules" are followed, I remain concerned that the results of this study cannot be reproduced.

Specifically taking the integrated moisture flux vectors in Fig. 3a as an example, some

of the moist flux vectors enter the study region from the SW, suggesting that moisture over the Lake Basin is a confluence of westerly and southwesterly advection. Is the westerly path considered dominant because it arrives at the "center" of the red box? It is not clear to me what justifies its selection as the dominant pathway influencing precipitation isotope ratios.

The following description also remains ambiguous: "The source regions of water vapor were determined based on the conditions for the formation of air masses, which need to form on a uniform underlying surface and possess stably in terms of isotopic, thermodynamic, and dynamic properties as well as circulation condition, typically located over vast land and ocean regions (Smirnov and Moore, 1999)". There is not enough guidance in this statement for another research team to try to replicate finding these source regions. If the paper is simply identifying source regions qualitatively by looking at where the vertically integrated moisture flux appears to originate from in the figures, then it needs to state this.

Response: We appreciated the helpful comments from the reviewer and provided a more transparent explanation of our methodology: "The water vapor transport paths were determined by identifying systematic vapor currents in the $Q$ field, which need to have the same directionality and draw the path along the central axis of the vapor currents. Our identified water vapor transport paths are conceptually similar to atmospheric rivers, which are long, narrow corridors of strong horizontal water vapor transport typically associated with low-level jets (Ralph et al., 2017; Payne et al., 2020). However, our paths are derived from the climatological mean state (multi-year monthly averages) rather than short-term events, which is a key distinction from atmospheric rivers that generally last for a few days to a week (Dettinger et al., 2013). The source regions of water vapor were determined based on the conditions for the formation of air masses, which need to form on a uniform underlying surface and possess stability and similarity in terms of isotopic, thermodynamic, and dynamic properties as well as circulation conditions (Smirnov and Moore, 1999). These regions are typically located over vast land and ocean areas and serve as the starting points of the water vapor transport paths. Although there is some empiricism and

subjectivity, there is no explicit algorithm to determine the exact water vapor transport path and source regions of water vapor. However, this approach is based on certain criteria and ensures that we capture the systematic vapor currents that have the most significant influence on the local precipitation and its isotopic composition, and it provides a reasonable basis for identifying the dominant vapor transport directions and the primary sources of water vapor contributing to precipitation in the Dongting Lake Basin" (Line 395-415).

**For the comment "Is the westerly path considered dominant because it arrives at the "center" of the red box? It is not clear to me what justifies its selection as the dominant pathway influencing precipitation isotope ratios".** In our study, the identification of the dominant water vapor transport path is based on the vertical integral of water vapor flux (i.e., the $Q$ field), with the Dongting Lake Basin (represented by the red box) as the endpoint. Specifically, the vector cluster of the water vapor flux (i.e., the $Q$) directed towards the Dongting Lake Basin delineates the path of water vapor transport. This approach allows us to identify the systematic vapor currents that converge towards the basin, thereby influencing the local precipitation and its isotopic composition. The westerly path is considered dominant because it represents the primary direction of water vapor convergence towards the Dongting Lake Basin, as indicated by the strongest and most consistent flux vectors in the $Q$ field. This path is not simply chosen because it arrives at the "center" of the red box, but rather because it reflects the dominant atmospheric circulation pattern influencing the region during the respective season. The selection of this path is based on the physical significance of the $Q$ field, which integrates both the magnitude and direction of water vapor transport. Therefore, we added the relevant discussion in the manuscript: "In the $Q$ field (Fig. 3a), regarding the Dongting Lake Basin (represented by the red box) as the endpoint, the vector cluster of the vertical integral of water vapor flux (i.e., the $Q$) directed towards the Dongting Lake Basin delineates the path of water vapor transport in January (black arrow lines in Fig. 3)" (Line 392-395) and "The westerly path is considered dominant because it represents the primary direction of water vapor convergence towards the Dongting Lake Basin, as indicated by the

strongest and most consistent flux vectors in the $Q$ field. The selection of this path is based on the physical significance of the $Q$ field, which integrates both the magnitude and direction of water vapor transport and reflects the dominant atmospheric circulation pattern influencing the region in this season" (Line 416-421)

**For the comments "Would we expect the transport paths identified in this work to agree with sources identified by tags? Would they agree with particle dispersion models or back trajectory models?"**, we believe that the water vapor transport paths should be primarily determined by the $Q$ field (the vertical integral of water vapor flux), which reflects the direction and magnitude of water vapor transport in the atmosphere. This approach is distinct from particle dispersion models or back-trajectory models, which focus on the movement of air particles. As we discussed in the manuscript: "The water vapor flux $Q$ reflects the direction and magnitude of water vapor transport in the atmosphere, while wind $V$ reflects the direction and magnitude of the movement of air particles in the atmosphere (Feng et al., 2009; Zhang et al., 2012; Zhang et al., 2016). There are both differences and connections between the two factors. Water vapor is transported by wind, and the wind carries water vapor from one place to another, and the directions of water vapor and wind may be consistent, inconsistent, or even opposite. In the East Asian monsoon region, the prevailing wind direction during the summer monsoon period is generally consistent with the average water vapor transport direction, both being southwest or southeast direction (Barker, et al., 2015; Wu et al., 2015; Tang et al., 2015). In this study, the water vapor transport path was consistent with the prevailing wind direction in June (Figs. 1a and 9). However, during the winter monsoon period, the prevailing wind direction may not be consistent with the average transport direction of water vapor—that is, the prevailing wind direction in January was northwest or northeast direction, while the average transport direction of water vapor in this period was southwest or southeast direction (Figs. 1a and 3)" (Line 853-867).

Furthermore, we provided an example from a weather-scale analysis to illustrate this point, as demonstrated in the manuscript: "Previous studies have shown that the most common weather systems and most precipitation events in the East Asian monsoon

region are caused by cold fronts resulting from the interaction of warm and cold air masses (Chen et al., 2020). According to classical meteorological theory (Zhou et al., 1997), in a cold front system, there appears the wind from the southwest direction blows ahead of the front, and a northwest wind blows behind the front as shown in the schematic diagram in Fig. 12a. Warm and moist air from low latitudes lifts along the front and leads to rainfall, while cold and dry air from high latitudes moves southward beneath the front and lifts the warm and moist air. At different heights and positions, the directions of air particle movement and water vapor transport are different. For example, at point A located above the warm and moist air side of the cold front surface, both air particles and water vapor are transported by southwest wind. At point C located below the cold, dry air side of the cold front surface, both air particles and water vapor are transported by northwest wind. However, at point B located within the front zone, the wind direction and speed are uncertain (Fig. 12b). Specifically, this front zone marked the transition from a warm air mass to a cold one, or vice versa, where meteorological factors have undergone rapid changes. Mixing between cold and warm advected air could occur within this zone, manifesting as a shear zone in wind fields, or as alternating southerly and northerly winds. Therefore, the dominant wind directions may not always align with the average water vapor transport direction, especially in frontal weather systems that dominate precipitation in the East Asian monsoon region.

[Figure]

Fig. 12 Schematic diagram of a cold front system in East Asia (based on Zhou et al., 1997)" (Line 868-897).

**In summary, while particle dispersion models and back-trajectory models provide valuable insights into air particle movement, they may not fully capture the complexities of water vapor transport, which is influenced by both large-scale atmospheric circulation and local weather systems. Our approach, based on the $Q$ field, aims to address these complexities by focusing directly on the transport of water vapor rather than relying solely on wind direction**. We hope this explanation clarifies our methodology and addresses your concerns. In addressing the potential discrepancies between our identified water vapor transport paths and those derived from particle dispersion models or back-trajectory models, it is important to emphasize the fundamental differences in the approaches used, thus we added the relevant demonstration in the manuscript: "Our study focuses on the $Q$ field (the vertical integral of water vapor flux) to determine water vapor transport paths, which directly reflects the direction and magnitude of water vapor transport in the atmosphere. This method is distinct from particle dispersion models or back-trajectory models, which are based on the movement of air particles (wind direction). As discussed earlier, water vapor is transported by wind, but the directions of water vapor transport and wind may not always align, especially in complex weather systems such as cold fronts" (Line 888-894).

2. Transport pathways during non-precipitating and precipitating conditions are not differentiated. I remain concerned that the analysis draws conclusions about transport pathways influencing precipitation isotope ratios using climatological transport characteristics representative of all weather conditions, regardless of whether or not precipitation occurs. The response-to-reviewer file suggests that another study (Xiao et al. 2024, DOI: 10.1175/JHM-D-23-0084.1) has already looked at conditions during precipitation days only; however, Xiao et al. evaluate the effect of precipitation type on precipitation isotope ratios. That paper does not look at transport pathways affecting precipitation over Dongting Lake Basin. I strongly recommend testing whether the results of this study are insensitive to whether or not non-precipitating days are included in the determination of transport pathways. If other studies have

already shown that shifts in transport pathways do not occur when precipitation is observed in Dongting Lake Basin, then citing those works would also address my concern satisfactorily.

Response: Thank you for raising this important point regarding the differentiation between transport pathways during precipitating and non-precipitating conditions.

In our study, **we analyze the climatological mean state of water vapor transport over four months (January, April, June, and October) from 1979 to 2017**, using monthly averages of vertical integral water vapor flux $Q$, precipitation amount $P$, and isotopic compositions of precipitation and water vapor. Our focus is on the general transport characteristics and their influence on precipitation in the Dongting Lake Basin, rather than individual events. This approach is justified because $Q$ is directly related to $P$, with low $Q$ values corresponding to low precipitation amounts in January and October (Figs. 4 and 10). Thus, we do not differentiate between precipitating and non-precipitating days. **Our previous study (Xiao et al. 2024) examined daily-scale variations and the relationship between precipitation isotopes and precipitation types during specific events, requiring differentiation between precipitating and non-precipitating days**. This complements our current analysis by focusing on individual events, while our manuscript examines the broader climatological context.

To address your concern, we have clarified in the manuscript that our analysis is based on the climatological mean state and does not differentiate between precipitating and non-precipitating days: "In this study, we focus on the climatological mean state of water vapor transport and its influence on precipitation in the Dongting Lake Basin over four representative months (January, April, June, and October, which representing winter, spring, summer, and autumn, respectively) from 1979 to 2017. Our analysis is based on the monthly average values of vertical integral water vapor flux $Q$, precipitation amount $P$, and isotopic compositions of water vapor and precipitation. This approach reflects the general characteristics of water vapor transport paths and their influence on precipitation for each month, rather than focusing on individual precipitation events. The magnitude of water vapor flux $Q$ is directly related to precipitation amounts $P$, and thus, our analysis does not require

Furthermore…

3. It is not clear what height the isotope ratios in vapor represent in the text and figures. Are these from a specific level? Or do they represent a mass integration of the tropospheric column?

Response: **In the isoGSM2, the water vapor isotope ratios (such as $\delta^{18}O_v$ and $\delta^2H_v$) are indeed derived from an integration throughout the entire atmospheric column**. This approach provides a comprehensive representation of the isotopic composition of water vapor across different levels, rather than focusing on a specific level. The integration accounts for the vertical distribution of water vapor and its isotopes, capturing the overall influence of atmospheric processes on the isotopic signatures. We apologize for the unclear statement and revised the relevant descriptions to "The water stable isotope simulation data used in this study are from isoGSM2 (January 1979 to December 2017, totaling 468 months), including monthly precipitation amount ($P$, mm), stable isotopes ($\delta^2H$ and $\delta^{18}O$) in the precipitation ($\delta^2H_p$, $\delta^{18}O_p$), the vertical integral of water vapor isotopes ($\delta^2H_v$, and $\delta^{18}O_v$) of 17 pressure levels from 1000 hPa to 10 hPa, and the calculated deuterium excess in water vapor and precipitation ($d_v$ and $d_p$)" (Line 263-268).

It is difficult to evaluate the study in much more detail without the aforementioned issues being clarified. That said, there are at least a few places that I will flag (minor comments, which build off concerns from my first review of this work):

1. Figure 6 is accompanied by a statement I flagged earlier: "following the variation rule of deuterium excess during water vapor transport—that is, as the rainout effect progressed, the heavier isotopes preferentially left the water vapor parcel…" If heavier isotopes are preferentially leaving the parcel, why is the isotope ratio of

precipitation increasing? This is inconsistent with the significant dehydration that would be required to raise the d significantly. (Also, it's not clear to me what the x-axis represents in this figure or where exactly the Lake Basin lies.)

Response: We appreciate the reviewer for pointing out the inconsistency in our explanation related to Fig. 6. We apologize for the incomplete description of our results, which may have led to confusion. The reviewer is correct that during the precipitation process, heavier isotopes preferentially leave the water vapor parcel or cloud due to the rainout effect. This process would typically result in a depletion of the remaining water vapor isotopes. However, **the increase in the isotope ratio of precipitation observed in our study can be attributed to the influence of additional factors. Specifically, the increase in precipitation isotope values may be related to the horizontal convergence of more enriched water vapor, which counteracts the depletion effect caused by rainout**. As shown in Figure 6a, before entering the Dongting Lake Basin along Path II, there is a significant increase in both the water vapor flux ($Q$) and precipitation amount ($P$). This suggests that the horizontal convergence of enriched surrounding water vapor plays a crucial role in influencing the isotopic composition of precipitation. The enriched water vapor from the surrounding water vapor contributes to the overall increase in the isotope ratios of both water vapor and precipitation, despite the ongoing rainout effect.

We revise the manuscript to provide a clearer and more comprehensive explanation of these processes: "Moreover, there was a large increase in both the $Q$ and $P$ before entering the Dongting Lake Basin along Path II, and the increase in the isotope ratio of precipitation may seem inconsistent with the rainout effect, which typically leads to the depletion of heavier isotopes in the remaining water vapor. However, this increase can be attributed to the horizontal convergence of more enriched water vapor. Path II showed a significant increase in both the $Q$ and $P$ before entering the Dongting Lake Basin (Fig. 6a). This suggested that the enriched water vapor from the surrounding water vapor contributed to the overall increase in the $\delta^{18}O_v$ and $\delta^{18}O_p$. The influx of this enriched water vapor counteracts the depletion effect caused by rainout, resulting in the observed increase in precipitation isotope values" (Line 538-547).

**Moreover, for the comment "Also, it's not clear to me what the x-axis represents in this figure or where exactly the Lake Basin lies", we have revised the figure caption and main text to provide clearer explanations**: "Fig. 4 Mean variations of $Q$ (a), $P$ (b), $\delta^{18}O_v$ (c), $\delta^{18}O_p$ (d), $d_v$ (e), and $d_p$ (f) along the vapor transport path in January. The numbers at the lower axis represent the serial numbers corresponding to the grid points along the water vapor transport path from the source region to the Dongting Lake Basin, and the points along the path were selected at almost equal intervals to capture the variations of each factor, hereinafter the same" (Line 450-454) and "In analyzing the variations along the water vapor transport paths, we selected several points along the path from the source region to the Dongting Lake Basin, spaced at nearly equal intervals. Six series of factors at the grid points along the water vapor transport path were derived from each factor field in January, including the variations of $Q$, $P$, $\delta^{18}O_v$, $\delta^{18}O_p$, $d_v$, and $d_p$, these points allow us to examine the changes in these factors along the transport path. Moreover, the grid points along the water vapor transport path were identified on the central axis of the path and based on the principle of uniform distribution of the scatter points, and the factors at the grid points were obtained from these scatter points. Besides, the factors at the grid points along the water vapor transport path exhibit, in spatial terms, average characteristics of conditions over multiple years, and, in temporal terms, sequential characteristics of these factors along the water vapor transport path" (Line 430-441).

2. With regards to the two April source regions, the Discussion states: "Clearly, the oceanic air mass with low deuterium excess had a relatively more significant impact on the precipitation isotopes in April in the Dongting Lake Basin region." This is overstated, given that both "source" regions have identical isotope ratios and differ in d-excess by only 1 permil. Any number of processes could influence the isotopic composition of these air masses as they travel many hundreds of kilometers to the observation site. This example illustrates a tendency for the paper to draw rather strong conclusions from what is overall a very qualitative analysis of climatological output from a GCM.

Response: Thank you for your insightful comments and for highlighting the need for more clarity in our discussion. Regarding the two source regions in April, our conclusion that the oceanic air mass with low deuterium excess had a relatively more significant impact on the precipitation isotopes in the Dongting Lake Basin region is not solely based on the isotopic compositions at the source regions. As you correctly pointed out, the isotopic ratios at the source regions are nearly identical, and the difference in deuterium excess is minimal (only 1‰). This underscores the complexity of isotopic variations along the water vapor transport paths, where numerous processes can influence the isotopic composition of air masses over long distances, and such variations are a continuous process with cumulative effects. These processes include rainout, evaporation recharge, mixing with other water vapor sources, and changes in temperature and humidity, all of which contribute to the cumulative isotopic variations along the transport paths.

However, **our primary criterion for determining the relative impact of these transport paths is the magnitude of the water vapor flux $Q$ transported along each path**. Specifically, as shown in Fig. 6a, the water vapor flux $Q$ associated with Path II is significantly higher than that of Path I before reaching the Dongting Lake Basin. This larger flux indicates that Path II contributes more water vapor to the region, thereby having a more substantial influence on the local precipitation isotopes. Therefore, we concluded that the oceanic air mass with low deuterium excess (associated with Path II) had a more significant impact on the precipitation isotopes in the Dongting Lake Basin region in April.

We have revised the relevant sections of the manuscript to clarify this point and avoid any potential overstatement: "Besides, it is worth noting the complexity of isotopic variations along the water vapor transport paths, where numerous processes can influence the isotopic composition of air masses over long distances, and such variations are a continuous process with cumulative effects. These processes include rainout, evaporation recharge, mixing with other water vapor sources, and changes in temperature and humidity, all of which contribute to the cumulative isotopic variations along the transport paths. Therefore, the primary criterion for determining

the relative impact of these paths is the water vapor flux $Q$. As shown in Fig. 6a, the water vapor flux $Q$ associated with Path II is significantly higher than that of Path I before reaching the Dongting Lake Basin. This larger flux indicated that Path II contributed more water vapor to the Dongting Lake Basin, thereby having a more substantial influence on the local precipitation isotopes. Clearly, the oceanic air mass with low deuterium excess had a relatively more significant impact on the precipitation isotopes in April in the Dongting Lake Basin region" (Line 773-786).

3. Line 807-811: This doesn't sound right. D-excess in the vapor of air masses is usually positive under most conditions observed near Earth's surface.

Response: We apologize for the unclear statement and appreciate the opportunity to clarify. **What we intended to convey is that in oceanic air masses, the isotopic values of water vapor (such as $\delta^{18}O$ and $\delta^2H$) are relatively higher compared to those in continental air masses. At the same time, the deuterium excess in oceanic water vapor is typically lower than that in continental water vapor**. In contrast, continental air masses generally have more depleted isotopic values due to the influence of rainout and other fractionation processes during transport. Therefore, we agree with your comment that d-excess in the vapor of air masses is usually positive under most conditions near Earth's surface, which can be found in the Figs. 3e, 3f, 5e, 5f, 7e, 7f, 9e, and 9f. We revised these sentences to "In oceanic air masses, the isotopic values of water vapor are relatively higher compared to those in continental air masses, while the deuterium excess of water vapor is typically lower. Conversely, in continental air masses, water vapor isotopes are relatively depleted but the d-excess can be higher (Rozanski et al., 1993; Araguás-Araguás et al., 1998)" (Line 811-815).

4. Line 829 - I'm still not convinced "modificatory" is the right word. This suggests an air mass that is influencing something else, not an air mass that is being modified by a process. What is intended?

Response: We apologize for the unclear statement and appreciate the opportunity to clarify. The term "modificatory" was intended to describe an air mass that undergoes

modification during its movement, primarily due to interactions with the underlying surface. This includes changes in temperature, pressure, humidity, and other properties as the air mass travels across different regions. We understand your concern that the term might imply an air mass influencing something else rather than being modified itself. **To address this, we have replaced "modificatory" with a more appropriate term "modified" to better convey the intended meaning. We have also provided a clearer explanation in the main text to ensure that the concept is well understood**: "East Asia is primarily controlled by the "modified" air masses, which were commonly used to describe the properties (e.g., temperature, pressure, and humidity) of air masses that have changed as they move through different regions (Ding, 1990; Chang et al., 2012). For instance, as an air mass moves from a warm and moist region to a cooler and drier region, its temperature and humidity can change significantly. Similarly, interactions with topography and varying surface conditions can lead to modifications in the air mass properties. This concept is particularly relevant in regions like East Asia, where air masses can be significantly altered as they pass through different climatic zones. Whether cold and dry air masses moving southward or warm and moist air masses moving northward, the isotopic composition of water vapor in the "modified" air mass continues to become more negative, while the deuterium excess of water vapor continues to become more positive than the original air mass, following the variation rule of stable isotope and deuterium excess during water vapor transport (Vasil'chuk, 2014; Zhou et al., 2019; Xu et al., 2019; Jackisch et al., 2022). In this study, interactions between "modified" oceanic air mass and "modified" continental air mass result in the water vapor isotope in Region V that differed from oceanic air masses and continental air masses (Table 1; Fig. 11)" (Line 830-846).

5. As noted in my first review, the Abstract claims that "the variations in water stable isotopes along water vapor transport paths adhered to Rayleigh fractionation and water balance principles." However, this is never tested and thus remains conjecture. Consequently, I do not feel that Objective 3 ("reveal the mechanisms by which water

vapor sources and transport paths in the monsoon region influence precipitation amounts and isotopes") is achieved.

Response: We appreciate the helpful comments from the reviewer and editor. Following the comments and suggestions, we have revised the sentence in the Abstract "In different seasons, the variations in water stable isotopes along water vapor transport paths adhered to Rayleigh fractionation and water balance principles" to "In different seasons, the variations in water stable isotopes along water vapor transport paths show some agreement with Rayleigh fractionation and water balance principles, as reflected in the model simulations and observations" (Line 25-28).

You are correct that our study does not provide direct empirical evidence to verify that the variations in water stable isotopes along water vapor transport paths strictly adhere to Rayleigh fractionation and water balance principles. Instead, our findings show some agreement between the model (which incorporates Rayleigh fractionation and water balance principles) and the observations. It reveals that the isotopic abundance of precipitation is influenced by the rainout effect along the water vapor transport path, en route water vapor replenishment, and the isotopic compositions of the source region. This analysis adheres to both the water mass balance and the stable isotope balance. This suggests that the model is capable of capturing the general trends in isotopic variations, but we acknowledge that this does not constitute direct proof of the underlying mechanisms. We have revised the abstract and the relevant sections of the manuscript to clarify this point and avoid any misleading implications. Objective 3 was revised to "(3) determine the influences of different water vapor sources and transport paths on the regional precipitation amounts" (Line 158-160) to better conform to the content of our manuscript.

6. Finally, I recommended previously and continue to recommend that the authors use traditional notation for deuterium excess: dp in place of Ex_dp.

Response: **We followed the comments and revised Ex_dp and Ex_dv to the traditional notation for deuterium excess in the main text, figures, and tables**. For instance we revised the relevant descriptions of the isoGSM2 dataset to "The water

stable isotope simulation data used in this study are from isoGSM2 (January 1979 to December 2017, totaling 468 months), including monthly precipitation amount ($P$, mm), stable isotopes ($\delta^2H$ and $\delta^{18}O$) in the precipitation ($\delta^2H_p$, $\delta^{18}O_p$), vertical integral of water vapor isotopes ($\delta^2H_v$, and $\delta^{18}O_v$) of 17 pressure levels from 1000 hPa to 10 hPa, and the calculated deuterium excess in water vapor and precipitation ($d_v$ and $d_p$)" (Line 263-268).